# MLFM: Multi-Layered Feature Maps for Richer Language Understanding in Zero-Shot Semantic Navigation

## Abstract

Recent progress in large vision-language models has driven improvements in language-based semantic navigation, where an embodied agent must reach a target object described in natural language. Yet we still lack a clear, language-focused evaluation framework to test how well agents ground the words in their instructions. We address this gap by proposing *LangNav*, an open-vocabulary multi-object navigation dataset with natural language goal descriptions (e.g. '*go to the red short candle on the table*') and corresponding fine-grained linguistic annotations (e.g., attributes: color=red, size=short; relations: support=on). These labels enable systematic evaluation of language understanding. To evaluate on this setting, we extend multi-object navigation task setting to *Language-guided Multi-Object Navigation* (*LaMoN*), where the agent must find a sequence of goals specified using language. Furthermore, we propose *Multi-Layered Feature Map* (*MLFM*), a novel method that builds a queryable, multi-layered semantic map from pretrained vision-language features and proves effective for reasoning over fine-grained attributes and spatial relations in goal descriptions. Experiments on *LangNav* show that *MLFM* outperforms state-of-the-art zero-shot mapping-based navigation baselines.

## 1 Introduction

Semantic navigation is a rapidly growing sub-field in embodied AI (Deitke et al., 2022) where an agent is tasked with navigating to a target object described using either an object category (Anderson et al., 2018a; Batra et al., 2020; Yokoyama et al., 2024), image (Krantz et al., 2022) or natural language descriptions (Li et al., 2021; Taioli et al., 2025). Each of these target specifications poses unique challenges. Among them, natural language is the most natural way for humans to express goals, yet it is also inherently ambiguous and context-dependent. In this paper, we focus on natural language guided semantic navigation, where goals are specified using language descriptions, for example, '*Go to the white table lamp on the wooden corner table*' (Figure 1).

Prior efforts to build datasets with language descriptions (Khanna et al., 2024b) often rely on large vision-language models (VLMs) to extract attributes from images and large language models (LLMs) to merge them into sentences. However, VLMs frequently hallucinate attributes (Li et al., 2023b), especially in cluttered or out-of-distribution scenes, introducing noise that biases both agent learning and evaluation. To mitigate this, we propose the *LangNav* dataset derived from the Habitat Synthetic Scenes Dataset (HSSD) (Khanna et al., 2024a). LangNav uses ground-truth attributes for objects, with all descriptions manually validated to avoid VLM errors. Furthermore, it provides fine-grained linguistic annotations–capturing object attributes (e.g. color, size, material) and spatial relations (e.g. 'on', 'near', 'left')–to enable systematic evaluation of language understanding.

Vision-and-language navigation (VLN) (Anderson et al., 2018b; Ku et al., 2020; Raychaudhuri et al., 2025) differs from our setting in that it provides full route instructions (e.g., "*leave the bedroom, turn left, pass the painting, and stop at the sofa*"), which are mostly crowdsourced and rarely contain hallucinated attributes. By contrast, object navigation specifies only the goal object, coupling grounding with exploration. In real-world interaction, people rarely provide step-by-step paths; instead, they say, "*bring me the blue mug on the bedside table*". Goal-only language is therefore both *natural*

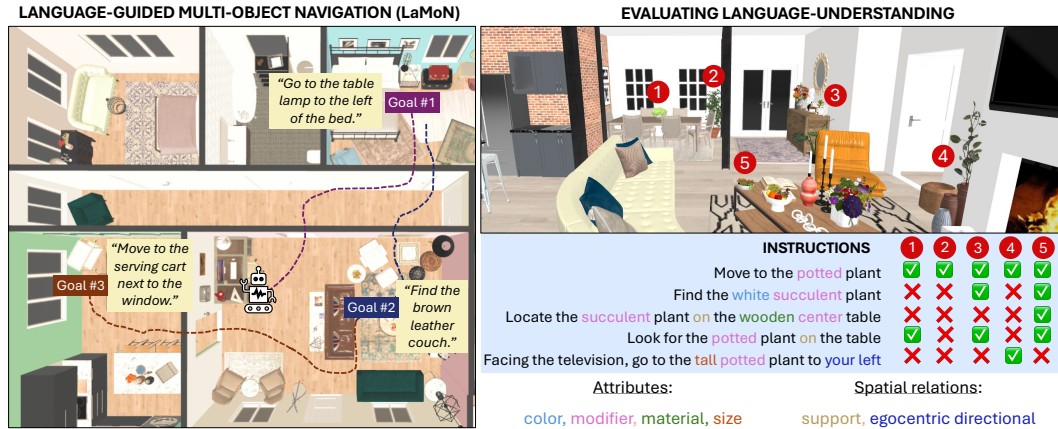

Figure 1: **Language-guided Multi-Object Navigation(LaMoN)** requires an agent to navigate to multiple goals, described using descriptions (left). We evaluate fine-grained language understanding by tagging each description with attributes (*color*) and spatial relations (*support*) (right). There may be multiple positive matches (objects matching all attributes and relations in the instruction) and the agent is scored correct if it stops at any (check marks a match and cross marks a non-match).

and *challenging*: the agent must explore unseen space, decide when it has found the correct object, and remember what it has already observed. Sequential tasks such as Multi-Object Navigation (MultiON) (Wani et al., 2020; Raychaudhuri et al., 2024) and GOAT-Bench (Chang et al., 2023; Khanna et al., 2024b) capture this difficulty by revealing each target only after the previous one is reached. We extend such task settings by proposing *Language-guided Multi-Object Navigation* (*LaMoN*) where each of three sequential goals is specified through a natural language description with different levels of specificity. For instance, in Figure 1, depending on the level of specificity, the description can match just one or multiple target objects. Success in LaMoN requires two key abilities: (*i*) building a semantic map memory while exploring, and (*ii*) querying that memory to reason about objects whose descriptions vary in granularity. To this end, we propose a novel method *Multi-Layered Feature Map* (*MLFM*) that constructs a multi-layered semantic map, leverages it to identify the goal object, semantically explores the environment and navigates via a path planner in an unseen environment.

In summary, our contributions are threefold: (i) *LangNav*: an open-vocabulary multi-object navigation dataset with natural language goal descriptions and fine-grained linguistic annotations, enabling systematic evaluation of language understanding; (ii) *LaMoN*: an extension of MultiON, where goals are specified using language descriptions at varying levels of specificity; (iii) *MLFM*: a multi-layer semantic-mapping approach that preserves rich object features and achieves improved performance over state-of-the-art zero-shot map-based semantic navigation baselines on the LangNav dataset[1].

## 2 RELATED WORK

**Semantic and Vision and Language Navigation.** In embodied AI, one of the basic semantic navigation tasks is the Object Navigation (ObjectNav), where an agent is required to navigate to an object specified by its category. While earlier benchmarks (Anderson et al., 2018a; Batra et al., 2020) have used a closed set of object categories, HM3D-OVON (Yokoyama et al., 2024) introduces a benchmark containing open-vocabulary categories to describe the goal. Multi-Object Navigation (MultiON) (Wani et al., 2020) extends this single object navigation to a multi-object setting where multiple objects are made available to the agent in sequence. MultiON requires the agent to remember objects observed in the past so as to come back if needed, for efficiency. Another type of semantic navigation is Instance-Nav (Krantz et al., 2022), where the agent is given an image of the goal object. While this task has its own challenges with respect to finding a specific object instance observed from a specific view angle, it does not reflect real-world scenarios where it is more likely for someone to describe an object of interest in words. A more natural semantic navigation is where

---

[1]The dataset and code will be released upon acceptance.

the goal object is described using natural language, a task called language-based InstanceNav (Li et al., 2021; Taioli et al., 2025). GOAT-Bench (Khanna et al., 2024b) was introduced to provide a benchmark on multi-object multi-modal navigation, combining different modalities (object category, image and language) to specify the goals. Compared to GOAT-Bench, where the language descriptions are generated via a VLM and contain significant errors, our LangNav dataset contains manually validated language descriptions at varying levels of specificity as well as annotations for fine-grained linguistic tags. Vision-and-Language Navigation (VLN) is another commonly studied task that differs fundamentally from our setting. VLN instructions explicitly describe the path the agent should follow, whereas in LangNav the agent receives only a goal description and must independently explore to locate the target. Recent advances in VLN include video-based VLM planners such as NaVid (Zhang et al., 2024) trained on VLN-CE (Krantz et al., 2020). Uni-NaVid Zhang et al. (2025a) unifies training on VLN-CE, ObjectNav, EQA, and human-following within a video-based VLA framework trained on 3.6M navigation samples. Other works address complementary dimensions: NaVILA (Cheng et al., 2025) proposes a two-level VLA architecture for legged robots; StreamVLN (Wei et al., 2025) introduces a SlowFast multi-modal streaming model for long-horizon VLN; and NavQ (Xu et al., 2025) learns a foresighted Q-model for instruction following. Additional related efforts include Learning Active Camera (Chen et al., 2022), which optimizes viewpoint selection for multi-object navigation, and MO-DDN (Wang et al., 2024b), which uses CLIP features and GPT-4–generated attribute descriptions for coarse-to-fine attribute-guided exploration.

**Semantic maps.** A widely used representation in semantic navigation is a semantic map that serves as a memory of observations that the agent has made throughout its navigation. It has been shown to improve efficiency in the longer-horizon MultiON task (Khanna et al., 2024b) by remembering objects of importance. While some works (Chaplot et al., 2020; Gervet et al., 2023; Zhang et al., 2025b) store explicit information such as object category in the map, more recent methods store implicit features in the map (Huang et al., 2023; Gadre et al., 2023; Chen et al., 2023), enabling open-vocabulary memory representation. These features are often extracted from a large VLM such as CLIP (Radford et al., 2021). VLMaps (Huang et al., 2023) uses pixel-level embeddings from LSeg (Li et al., 2022), while VLFM (Yokoyama et al., 2023) uses BLIP-2 (Li et al., 2023a) image features to build a 2D top-down map and compute semantic frontiers to explore the environment while performing navigation. Others build 3D maps (Zhang et al., 2023; Gu et al., 2024) which preserve geometric details of the objects and allow better reasoning, but are very costly to build and maintain, especially in long-horizon tasks. In this work, we propose a *multi-layer map* that serves as a middle-ground between a 2D and a 3D grid map and contains stacked layers of 2D maps, providing more granularity and flexibility than a 2D map but avoids the cost of a full 3D grid map.

**Evaluation benchmarks and datasets.** Current semantic navigation methods focus on task success, i.e. whether the agent is able to find a goal, but lack an evaluation framework to test language understanding. At the same time, in other fields such as 3D scene understanding and grounding, evaluating AI models against natural language understanding has gained a lot of interest. OpenLex3D (Kassab et al., 2025) proposes a benchmark that evaluates open-vocabulary scene representation methods by introducing new label annotations, such as synonyms, depictions, visual similarity, and clutter, in existing scene datasets. Eval3D (Duggal et al., 2025) proposes an evaluation tool to evaluate 3D generative models to assess their geometric and semantic consistencies. Agentbench (Liu et al., 2023) introduces a testbed to systematically evaluate LLMs acting as agents on their reasoning capabilities. ViGiL3D (Wang et al., 2024a) introduces a dataset for evaluating visual grounding methods with respect to a diverse set of language patterns, such as object attributes, relationships between objects, target references, etc. Motivated by these works, we introduce *LangNav*, a dataset containing fine-grained linguistic annotations to evaluate semantic navigation methods.

## 3 TASK AND DATASET

**Task.** Our *Language-guided Multi-Object Navigation* (*LaMoN*) task is an extension of the MultiON task (Wani et al., 2020; Raychaudhuri et al., 2024) and the GOAT-Bench task (Chang et al., 2023; Khanna et al., 2024b), where an agent is spawned in an unseen indoor environment and tasked with sequentially navigating to multiple goal objects. While MultiON uses a closed-vocabulary category name to describe each goal, GOAT-Bench uses multiple modalities–image, category and open-vocabulary language descriptions–to describe each goal. In contrast, we set up our task in an open-vocabulary setting, specifying each goal using *language descriptions with different levels*

Table 1: **Comparison with prior datasets.** LangNav contains open-vocabulary object+instance goals, annotated with fine-grained linguistic tags and language descriptions are manually verified and exist at different levels of specificity (*Specificity*), depending on which it can match with one or multiple correct target objects.

| Datasets | Task | ObjectNav goals | InstanceNav language goals | Open-vocab | Linguistic tags | No significant description errors | Specif-icity |
|---|---|---|---|---|---|---|---|
| OVMM(Yenamandra et al., 2023) | Manipulation | ✓ | ✗ | ✓ | ✗ | - | - |
| ObjectNav(Batra et al., 2020; Yadav et al., 2022) | Object nav | ✗ | ✗ | ✗ | ✗ | - | - |
| HM3D-OVON(Yokoyama et al., 2024) | Object nav | ✓ | ✗ | ✓ | ✗ | - | - |
| MultiON(Wani et al., 2020; Raychaudhuri et al., 2024) | MultiON | ✓ | ✗ | ✗ | ✗ | - | - |
| OneMap(Busch et al., 2025) | Object nav, MultiON | ✓ | ✗ | ✗ | ✗ | - | - |
| Goat-Bench(Khanna et al., 2024b) | Multi-(object+instance) nav | ✓ | ✓ | ✓ | ✗ | ✗ | ✗ |
| *LangNav* | Multi-(object+instance) nav | ✓ | ✓ | ✓ | ✓ | ✓ | ✓ |

*of specificity*. Each episode contains *three* goals, disclosed to the agent one at a time. Language descriptions in *LaMoN* describe objects with varying levels of specificity, e.g. 'go to the couch' vs. 'go to the black couch' vs. 'go to the black three piece L-shaped sectional couch'. This allows us to evaluate the agent's capability to understand coarse vs fine-grained linguistic cues in the descriptions. At every step, the agent takes as inputs egocentric RGB-D images, GPS and compass readings relative to the agent's starting pose in the episode, and the language instruction for the current goal $g_i$, The agent takes one of four actions: *move forward* by 25 cm, *turn left* by 30° or *turn right* by 30°, and *found*. A goal is successful if the agent generates *found* within 1.5 m of the goal object and under 500 steps. The episode continues even when the agent fails to navigate to one goal.

### 3.1 DATASET

In this section, we introduce the *LangNav* dataset that implements the LaMoN task. Among the prior datasets to evaluate various semantic navigation tasks, GOAT-Bench (Khanna et al., 2024b) is the closest to ours (Table 1). However, due to the use of a VLM during creation without manual verification, their language goals contain significant errors (see Appendix A.1).

**Evaluation focus.** LangNav is designed primarily as an evaluation benchmark: its goal is to measure how well existing agents generalize to fine-grained, spatially grounded language descriptions, rather than to provide supervision for training new models.

**Episode generation.** We use high-quality synthetic 3D scenes from HSSD (Khanna et al., 2024a). For each episode we draw a random navigable start pose for the agent and choose three goals from the objects in the scene. For each goal, we store the *language description*, *linguistic tags* along with the *viewpoints* or navigable positions around the goal (see Appendix A.1 for detail).

**Attribute descriptions.** We convert the ground-truth attributes available in HSSD into fluent object descriptions with GPT-4 (OpenAI, 2023). The model is prompted with in-context examples (Brown et al., 2020; Liu et al., 2021; Wu et al., 2023) to form a coherent natural language description. Next, we use GPT-4 to also tag every instruction with the linguistic cues (Wang et al., 2024a) and release the annotations along with the dataset: • **Color**—a color adjective in the description, e.g. 'the *brown* chair'. • **Size**—an explicit size cue, e.g. 'the *small* candle'. • **Texture**—a surface pattern or finish, e.g. '*knitted* bath mat'. • **State**—a transient state that could change, e.g. '*illuminated* makeup mirror'. • **Number**—a numeral in the description, e.g. 'the cabinet with *two* doors'. • **Material**—the substance an object is made of, e.g. '*wooden* shoe rack'. • **Modifier**—any other adjectival qualifier, e.g. 'the *easy* chair'.

**Spatial relation descriptions.** We programmatically extract spatial relationships between a pair of objects from the ground-truth 3D bounding boxes in HSSD, following methodologies similar to prior works in text-conditioned 3D scene generation (Chang et al., 2014; Tam et al., 2025) (see Appendix A.1). We extract the following spatial relations: • **Egocentric directional**–a cardinal direction relative to the agent's orientation, e.g. 'the plant to *your left*'. • **Allocentric**–a spatial relation relative to other objects independent to the agent's current pose and further sub-divided into: (a) **Support**–an explicit support relation, e.g. 'the potted plant *on* the console table'. (b) **Directional**–a direction-dependent spatial arrangement which is not a support relation, e.g. 'the mirror *above* the sink'. (c) **Proximity**–a relation encoding the notion of metric or qualitative closeness, e.g. 'the floor lamp *near* the bed' or 'the plant *next to* the couch'. (d) **Containment**–an inclusion relationship, e.g. 'the plush toy *inside* the crib'.

**Statistics.** We generate two splits for our dataset: validation and test, with distinct set of target objects and distinct scenes. The validation split contains 932 episodes, each with three goals, totaling 2796 goal descriptions spanning 20 scenes. The test split contains 855 episodes, each with three goals, totaling 2565 goal descriptions spanning 15 scenes (see Appendix A.1).

## 4 METHOD

We propose *Multi-Layered Feature Map* (*MLFM*), a zero-shot semantic navigation method that incrementally builds a semantic map by storing visual features from a pretrained vision-language model (Radford et al., 2021). At each step, MLFM queries the map by comparing stored features with the goal's language embedding, generating a similarity heatmap over cells. If a likely target cell is found, the agent navigates to it via a path planner; otherwise, it continues exploring using a semantic exploration method VLFM (Yokoyama et al., 2023).

**Two phase navigation (EAE-E).** For navigation, we introduce a two-phase method called *EAE-E*. At the beginning of an episode the map is still sparse and because it stores *features* rather than explicit class labels, its cosine scores can be noisy. An open-vocabulary detector, e.g. YOLO-World (Cheng et al., 2024), on the other hand, yields high-precision category predictions whenever the object is in view. Hence, early on the detector is the more reliable signal while the map mainly proposes candidate locations that must later be verified by the detector. As exploration continues and the map densifies, trust gradually shifts toward the map. Following this idea we introduce two-phase navigation: (*i*) **Explore-And-Exploit (EAE).** For the first $\mathcal{N}_e$ steps a location is accepted as the goal only if *both* the similarity map $\mathcal{S}$ and the detector agree, similar to consensus filtering (Busch et al., 2025). Upon agreement the agent moves to that cell via the path planner; otherwise it resumes exploration. (*ii*) **Exploit-only (E).** For the remaining $(\mathcal{N} - \mathcal{N}_e)$ steps it relies solely on the map, selecting the cell with highest similarity to thes goal description and navigates to it.

### 4.1 MAPPING

The motivation behind using a multi-layered map is that it allows us to capture the vertical context without incurring the cubic memory cost of a full 3D voxel grid. Though flattening everything into a single 2D plane is a commonly used approach, doing so merges features from stacked objects—small items vanish beneath larger ones—and discards the height cues needed for reasoning about spatial relations between objects. We therefore adopt a *multi-layer* top-down map that stores only a handful of horizontal slices: granular enough to separate objects at different heights, yet linear in memory and update cost. To this end, we build a multi-layer feature map, $\mathcal{M} \in \mathbb{R}^{L \times h \times w \times f_d}$ where $L$ is the number of layers, each containing a 2D top-down map $m \in \mathbb{R}^{h \times w \times f_d}$ and $f_d$ is the feature dimension stored at each map cell. $h$ and $w$ correspond to the ($X$ x $Y$) space of the physical environment, while $L$ corresponds to its discretized height dimension $Z$. Figure 2 presents a schematic of the map building process. Given the RGB observation $I_t \in \mathbb{R}^{H \times W \times 3}$ at each step $t$, we extract patch-wise features $Fp_t \in \mathbb{R}^{H \times W \times n_p \times f_d}$ using CLIP (Radford et al., 2021) image encoder from SED (Xie et al., 2024). We rely on a CLIP-based vision–language model to store features embedded in a joint image–text space, so that the map can be queried with natural language descriptions. Here $n_p$ is the number of patches. Using the depth observation $D_t \in \mathbb{R}^{H \times W}$ and the camera intrinsics, we then project the features $Fp_t$ onto a 3D point-cloud. Next we project the 3D points onto the 2D plane. For each point $p_i$ in the point-cloud at the 3D location $(x_i, y_i, z_i)$, we find the appropriate layer using the height $z_i$ of the point in the point-cloud, such that: $l_i = \left\lfloor \frac{h_i}{\Delta h} \right\rfloor$, where $l_i$ is the layer index and $\Delta h$ denotes the height range of each layer.

### 4.2 QUERYING

We implement three different querying techniques for MLFM to query the built map with the input goal description. Below we describe each technique.

**MLFM-vanilla.** This variant relies on the most basic querying technique to find the target on the multi-layer map by identifying candidates on the map where the cosine similarity between stored visual features and the query text is above a threshold and then selects the highest score as the target.

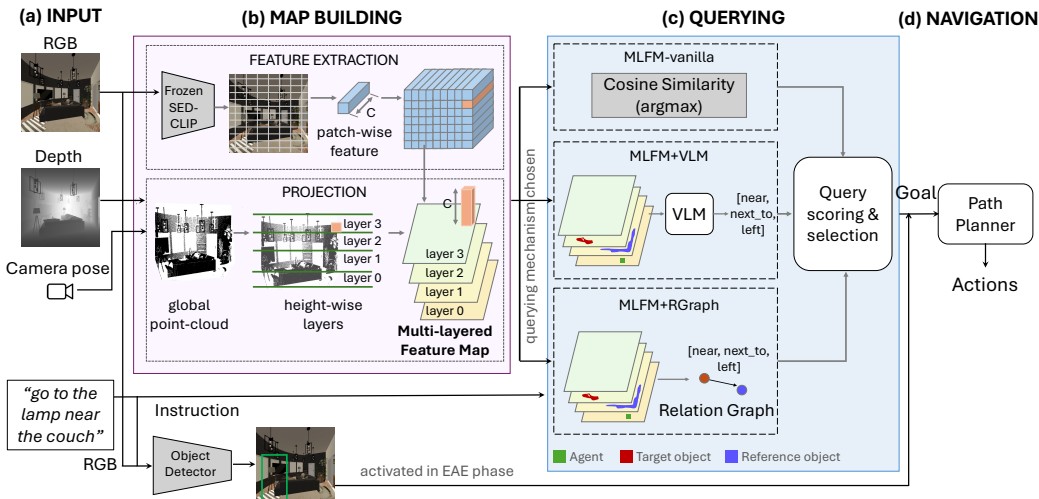

Figure 2: **Method.** (a) The agent takes as input the RGB image from which the map building (b) extracts learned visual embeddings and projects onto layers using the depth and camera pose inputs. The map is then queried based on the input instruction by employing one of three techniques (c)– *vanilla*, *VLM* or *RGraph*. Once the agent identifies a possible goal location, it navigates to it using a path planner (d). The agent activates the object detector as an additional signal during the initial phase (EAE) of the navigation.

**MLFM+VLM.** In this variant, we use a large vision-language model (VLM) to query the relationship between objects. We first find the most likely locations of the target object and the reference object by computing the highest similarity scores for each. We then color the target object with 'red' and the reference object with 'blue'. We additionally mark the agent location on the map with a 'green' square. We generate an image with image tiles corresponding to each layer of the map and indicate that tile 1 is at a lower height than tile 2, which is at a lower height than tile 3 and so on. We then prompt GPT-4 and ask it to infer the spatial relation between the 'red' object A and the 'blue' object B, from the agent's viewpoint: *"Infer the spatial relation between red A and blue B. If a green box is present, it indicates agent location. Decide relation from the agent's viewpoint."*. We compare the inferred relation to the query relation by encoding them with a CLIP text encoder and computing the cosine similarity. If the score is above a threshold, we consider the target as found.

**MLFM+RGraph.** In this variant we construct a *relation graph* (*RGraph*), with nodes representing objects and directed edges encoding spatial relations between object pairs (e.g., edge from A to B with label 'above' indicates 'A is above B'). Each node corresponds to a candidate location of either the target or the reference object, identified from the multi-layered map if its similarity score with the text query exceeds a threshold. Along with the object identity, we associate each node with the visual features extracted from its corresponding map region. Edges are introduced between two nodes if the objects are in close spatial proximity, determined by the Euclidean distance on the map. Each edge is labeled with one or more relations inferred from the object map locations, and stores the corresponding CLIP text features. Note that this supports multiple relations between A and B, since A can be near B and at the same time to the left of B. In this way, the RGraph explicitly captures not only objects but also pairwise relations grounded in spatial structure and allows the agent to disambiguate which specific object pair is referenced (e.g. 'the picture next to the cabinet' vs. 'the picture above the cabinet'). See Appendix A.2.3 for more details on the graph construction.

## 5 EXPERIMENTS

In this section, we compare MLFM with state-of-the-art zero-shot semantic navigation methods on our LangNav dataset.

**Implementation.** For all the baselines, we use an A* (Hart et al., 1968; Busch et al., 2025) planner. We use SED (Xie et al., 2024) patch features to build the multi-layer map and Yolo-World as an object detector in MLFM. For experiments reported in this section, our map has three layers and each 2D grid cell side maps to 6 cm in the physical space. Using 1×A40 GPU, it takes ∼8 hours for

evaluating our method on the full test split. For our experiments, we report the mean performance over five runs with random seeds and observe a standard deviation of $\pm 2.5$.

**Evaluation metrics.** We use standard navigation metrics, the success rate (*SR*) and the success weighted by inverse Path Length (*SPL*) (Anderson et al., 2018a) to measure whether the agent can reach a target object successfully and how well the agent trajectory matches the shortest path to the target respectively. More specifically, we report SR against each of the linguistic features–attributes (*color, size*, etc.) and spatial relations (*support, egocentric directional*, etc.).

**Baselines.** We focus on understanding how an explicit semantic map helps agents solve the La-MoNtask on the LangNav dataset, especially when the goal descriptions vary in linguistic attributes and spatial relations. Accordingly, we compare MLFM with state-of-the-art *mapping-based* zero-shot navigation methods: **VLMaps** (Huang et al., 2023), **VLFM** (Yokoyama et al., 2023), **OneMap** (Busch et al., 2025) and **MOPA** (Raychaudhuri et al., 2024) (see Appendix A.3). We have two additional baselines, **VLFM-v2** and **OneMap-v2**, where we adapt the two phase-navigation EAE-E (Section 4), explore-and-exploit (EAE) followed by exploit-only (E) phases, for the two baselines, VLFM and OneMap. We then compare with a baseline **EgoImageMap+VLM**, where we store egocentric RGB images on the map, similar to Modular GOAT (Chang et al., 2023; Khanna et al., 2024b). We then use the two phase-navigation (EAE-E) and query GPT-5 with the stored images to select the goal. We additionally include two recent video-based VLM baselines, **NaVid** (Zhang et al., 2024) and **Uni-NaVid** (Zhang et al., 2025a). NaVid is trained on VLN-CE route-following, while Uni-NaVid unifies VLN-CE, ObjectNav, EQA, and human-following within a single video-based VLA framework. Both methods are evaluated on our task as non-mapping-based reference points; however, it is worth noting that they are fine-tuned on navigation datasets and therefore cannot be fully considered zero-shot.

## 5.1 RESULTS ON LANGNAV

Below we discuss agent performance on various aspects of language understanding. Table 2 shows that MLFM variants (rows 10-12) outperform the 2D map based baselines (rows 1-7) on overall SR and SPL, highlighting the effectiveness of the granular multi-layered representation over 2D top-down maps. This applies to fine-grained attribute understanding (with few exceptions), spatial understanding, as well as descriptions containing only goal category (+3.9% gain in row 10 over 6).

**Multi-layered maps capture fine-grained linguistic cues better than 2D maps.** Table 2 shows that MLFM significantly outperforms the baselines (rows 10-12 vs. 1-7) in attribute understanding, especially on attributes–color, number, material and modifier–by +3.8%, +7.2%, +51.6% and +2.8% respectively (row 10 vs. 6). This indicates that the layered map represents visual features better than a 2D map, where the projected features get aggregated on a single layer, making it hard to distinguish intricate details about objects. Moreover, MLFM+RGraph outperforms MLFM-vanilla and MLFM+VLM on color, material and modifier attributes. However, MLFM fails to identify texture of objects. This is mainly the limitation of the feature extractor (see ablation in Section 5.3 and the discussion in Appendix A.4). MLFM also fell short of OneMap-v2 on descriptions with the state attribute (see analysis is Section 5.2).

**Two-phase navigation is effective in long-horizon tasks.** Poor performance in VLFM can be attributed to the reliance on the built map, even during the initial stages of navigation when the map may not contain the target object. This is evident from the improved performance of VLFM-v2 (row 5), where we adapt the two-phase navigation, over the vanilla VLFM (row 2), even when both use 2D maps. We observe a similar trend in OneMap (rows 6 vs. 4). This shows that our two-phase navigation (EAE-E) approach is extremely effective when dealing with longer-horizon tasks.

**Multi-layered maps capture spatial relations better than 2D maps.** Table 2 shows that *MLFM-vanilla* with its basic querying mechanism outperforms VLFM-v2 and OneMap-v2 (rows 10 vs. 5-6) across all spatial relation types. This reaffirms that the multi-layer map not only helps in attribute understanding, but the granular representation also helps in spatial understanding.

**Spatial understanding benefits from enhanced representation and querying.** Both *MLFM+VLM* (row 11) and *MLFM+RGraph* (row 12), equipped with enhanced querying mechanisms, outperform *MLFM-vanilla* across all relation types, with *MLFM+RGraph* achieving the

Table 2: **Performance.** MLFM+RGraph outperforms other mapping-based zero-shot methods on overall metrics, most attributes and spatial relations. The MLFM variants do best on 'cat' and on 'color', 'num', 'mat' and 'mod' attributes. *: denotes methods fine-tuned on navigation tasks. Note-*cat*:category, *tex*:texture, *num*:number, *mat*:material, *mod*:modifier, *ego-dir*:egocentric directional, *allo-dir*:allocentric directional, *supp*:support, *prox*:proximity, *cont*:containment.

| Methods | Overall | | | Attribute understanding | | | | | | | Spatial relations understanding | | | | |
|---|---|---|---|---|---|---|---|---|---|---|---|---|---|---|---|
| | SR↑ | SPL↑ | cat | color | size | tex | num | mat | state | mod | ego-dir | allo-dir | supp | prox | cont |
| 1) VLMaps | 5.0 | 1.2 | 3.1 | 1.4 | 0.0 | 0.0 | 0.0 | 0.0 | 0.0 | 3.0 | 0.0 | 1.0 | 0.6 | 3.7 | 0.0 |
| 2) VLFM | 7.1 | 2.6 | 2.4 | 1.6 | 0.0 | 0.0 | 0.0 | 0.0 | 0.0 | 3.3 | 0.4 | 1.6 | 1.3 | 6.9 | 2.1 |
| 3) MOPA | 6.3 | 4.4 | 2.8 | 2.6 | 3.6 | 0.0 | 0.0 | 0.0 | 0.0 | 4.2 | 0.0 | 2.1 | 1.4 | 4.1 | 2.0 |
| 4) OneMap | 15.3 | 7.1 | 3.7 | 4.7 | 3.9 | 3.0 | 2.3 | 3.1 | 0.0 | 6.7 | 2.1 | 4.1 | 10.7 | 5.5 | 5.7 |
| 5) VLFM-v2 | 24.8 | 9.7 | 28.2 | 11.5 | 33.3 | 0.0 | 7.1 | 43.1 | 28.6 | 17.7 | 0.0 | 2.8 | 12.9 | 4.2 | 3.3 |
| 6) OneMap-v2 | 26.7 | 10.1 | 29.1 | 26.9 | **41.7** | 0.0 | 7.1 | 0.0 | **57.1** | 21.1 | 2.1 | 4.7 | 22.1 | 5.9 | 6.5 |
| 7) EgoImageMap+VLM | 35.1 | 10.9 | 33.7 | 34.8 | 39.5 | **33.3** | 11.7 | 53.1 | 17.8 | 35.0 | 34.6 | 20.2 | 33.3 | 22.2 | 17.5 |
| 8) NaVid* | 36.1 | 12.7 | 33.1 | 35.2 | 11.4 | 0.0 | 10.7 | 50.8 | 30.4 | 34.5 | 31.6 | 15.8 | 24.0 | 25.4 | 17.2 |
| 9) UniNaVid* | **40.0** | 13.3 | **38.4** | **37.1** | 20.7 | 11.1 | 12.4 | **55.1** | 41.0 | **39.2** | 35.2 | 14.3 | 29.9 | 28.0 | 20.8 |
| 10) MLFM-vanilla | 28.8 | 10.3 | 33.0 | 30.7 | **41.7** | 0.0 | **14.3** | 51.6 | 14.3 | 23.9 | 16.7 | 12.5 | 20.4 | 14.1 | 14.3 |
| 11) MLFM+VLM | 33.3 | 10.7 | 33.3 | 34.5 | **41.7** | 0.0 | **14.3** | 51.9 | 14.3 | 31.3 | 17.1 | 19.3 | 21.2 | 16.9 | 14.7 |
| 12) MLFM+RGraph | 39.5 | **14.9** | 33.6 | 34.9 | **41.7** | 0.0 | **14.3** | 53.8 | 14.3 | 37.6 | **37.7** | **20.7** | **34.2** | **32.5** | **21.1** |

best performance. This underscores that spatial relationships can be effectively modeled as graphs, allowing for better reasoning capabilities.

**VLMs reason better on raw egocentric images than top-down abstract projections.** *EgoImageMap+VLM* (row 7) achieves comparable performance as our best performing model, and outperforms *MLFM+VLM*. This indicates that the VLM is able to reason over egocentric RGB images better than top-down projections of image features (see Section 5.2). Moreover, it achieves the best results on descriptions with only goal category and with texture attribute.

**Video-based VLM agents offer a complementary reference point.** Table 2 also reports two recent video-based navigation models, *NaVid* and *Uni-NaVid* (rows 8–9), which operate without explicit mapping. NaVid, trained only on VLN-CE route-following, performs noticeably below our mapping-based methods (36.1% SR), illustrating the difficulty of transferring route-conditioned behavior to exploration-style object navigation. Uni-NaVid, trained jointly on VLN-CE, ObjectNav, EQA and human-following, attains an SR comparable to MLFM+RGraph (40.0% vs. 39.5%). However, the methods diverge in two important ways. First, MLFM+RGraph achieves higher trajectory efficiency (14.9 vs. 13.3 SPL) and consistently outperforms both NaVid and Uni-NaVid on categories requiring structured spatial reasoning, such as number, support, containment and egocentric direction, where the explicit multi-layer map provides stronger geometric grounding. Second, Uni-NaVid (and to a lesser extent NaVid) performs slightly better on attributes dominated by raw detection quality, including color, basic category and texture; these differences primarily reflect the strength of the underlying detector rather than the effectiveness of the navigation policy itself. Overall, the comparison highlights the complementary strengths of video-based VLMs and structured mapping, with MLFM+RGraph providing more reliable fine-grained grounding while remaining competitive in overall success.

## 5.2 FAILURE ANALYSIS

**MLFM vs. OneMap-v2.** To understand why OneMap-v2 performs better than MLFM-vanilla for descriptions with *state* attribute, we compute the percentage of times the agent makes a wrong detection or identifies the wrong goal on the map or runs out of time. First, we observe that there are more wrong detections (71.4% in MLFM-vanilla vs. 28.6% in OneMap-v2) than wrong goals on the map (14.3% in both). This indicates that performance could be improved by using a better object detector (confirmed by ablation in Section 5.3). Second, MLFM-vanilla and OneMap-v2 have the same wrong goal on map percentage (14.3%), indicating that this error is not due to using multi-layer map instead of 2D map. Moreover, an ablation in Section 5.3 indicates that the performance in *state* attribute improves by using image-level features instead of patch-level features.

**Overall MLFM failure.** Performing a similar failure analysis on the overall performance of MLFM-vanilla, we observe that 31.7% of the error cases are due to wrong detection and 17.0% are due to wrong goal identification on the map. The remaining 48.3% is due to the agent running out of time budget while exploring and 3% is where the agent identifies the goal but fails to reach it on time.

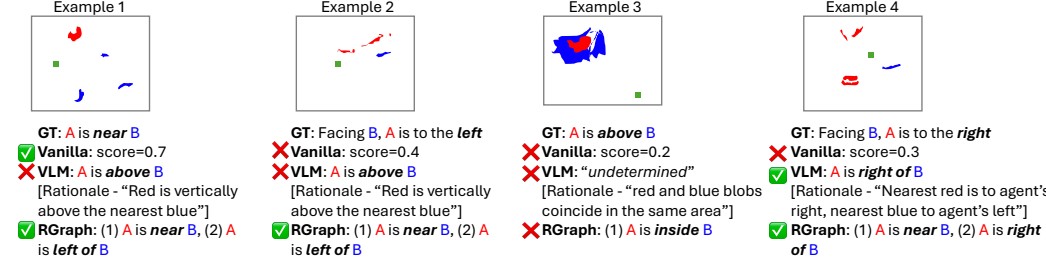

**GT**: A is *near* B
✅**Vanilla**: score=0.7
❌**VLM**: A is *above* B
[Rationale – "Red is vertically above the nearest blue"]
✅**RGraph**: (1) A is *near* B, (2) A is *left of* B

**GT**: Facing B, A is to the *left*
❌**Vanilla**: score=0.4
❌**VLM**: A is *above* B
[Rationale – "Red is vertically above the nearest blue"]
✅**RGraph**: (1) A is *near* B, (2) A is *left of* B

**GT**: A is *above* B
❌**Vanilla**: score=0.2
❌**VLM**: "*undetermined*"
[Rationale – "red and blue blobs coincide in the same area"]
❌**RGraph**: (1) A is *inside* B

**GT**: Facing B, A is to the *right*
❌**Vanilla**: score=0.3
❌**VLM**: A is *right of* B
[Rationale – "Nearest red is to agent's right, nearest blue to agent's left"]
✅**RGraph**: (1) A is *near* B, (2) A is *right of* B

\* for *vanilla* we use a threshold score 0.6, above which it is considered positive match

Figure 3: Comparisons showing that *VLM* struggles to reason on projected abstract features, often interpreting them as egocentric views (example 1). *RGraph* struggles distinguishing 'inside' from 'above' when both objects are projected onto the same map layer (example 3).

**MLFM+VLM vs. MLFM+RGraph.** We find that the primary source of failure for *MLFM+VLM* arises from the bias of VLMs, which are trained on web-scale photo and video corpora dominated by egocentric perspectives. As a result, they infer spatial relations more reliably from egocentric images than from top-down maps. For instance, when presented with a top-down projection where A's red blob appears above B's blue blob, the VLM tends to predict 'A is above B', mirroring how it would interpret an egocentric image, rather than the intended relation A is 'near' or 'next to' B (see Figure 3). *MLFM+RGraph*, on the other hand, might fail to differentiate between 'inside', 'above', 'below' and 'on top of' if both the objects are projected on the same map layer, depending on the number of layers in the multi-layered map.

## 5.3 ABLATIONS

To gain more insights into why MLFM fails to understand attributes like *texture* and *state*, we perform ablations (Table 3) with open-vocabulary feature extractors and object detectors.

**Patch vs. image vs. pixel features.** When compared with their image-level counterparts (rows 1–2 vs. 3–4), SED-CLIP and BLIP-2 patches generally perform better on attribute understanding (with a few exceptions). This can be attributed to the fact that when we store the same global feature vector in every layer (CLIP or BLIP image-level), the map gains no additional information. This is evident for *color*, *size*, *material*, and *modifier* cues ($+26.9\%$, $+25.0\%$, $+16.3\%$, $+3.1\%$, respective gains in row 2 over 4). Image features, on the other hand, edge out patches on *number* and *state* attributes (rows 2 vs. 4), suggesting that a holistic view occasionally helps when counting instances or detecting object states. Pixel features from LSeg perform worst overall (row 5), indicating that overly fine-grained features hinder object-level reasoning and dilute broader context.

**BLIP-2 vs. CLIP.** BLIP-2 patches outperforms SED-CLIP patches on the *color* ($+3.9\%$) and *modifier* ($+1.0\%$) attributes (row 2 vs. 1). Additionally, BLIP-2 image outperforms CLIP on *texture* ($+20.0\%$) and *number* ($+21.4\%$) comprehension (row 4 vs. 3). These findings confirm BLIP-2's stronger vision-language alignment for such properties.

Table 3: **Ablations.** Patch outperforms image followed by pixel features. However, the choice of feature extractor and object detector might influence understanding of certain attribute types.

| Object detector | Features | Type | Attribute understanding | | | | | | |
|---|---|---|---|---|---|---|---|---|---|
| | | | color | size | texture | number | material | state | modifier |
| 1) YOLO-World (Cheng et al., 2024) | SED (Xie et al., 2024) | patch | 30.7 | 41.7 | 0.0 | 14.3 | **51.6** | 14.3 | 23.9 |
| 2) | BLIP-2 (Li et al., 2023a) | patch | **34.6** | 41.7 | 20.0 | 14.3 | 16.3 | 14.3 | 24.9 |
| 3) | CLIP (Radford et al., 2021) | image | 11.5 | 8.3 | 0.0 | 0.0 | 0.0 | 28.6 | 17.1 |
| 4) | BLIP-2 (Li et al., 2023a) | image | 7.7 | 16.7 | **20.0** | **21.4** | 0.0 | 28.6 | 21.8 |
| 5) | LSeg (Li et al., 2022) | pixel | 3.8 | 0.0 | 0.0 | 7.1 | 0.0 | 14.3 | 7.1 |
| 6) Grounding-DINO (Liu et al., 2024) | SED (Xie et al., 2024) | patch | 26.9 | **49.7** | 0.0 | 0.0 | 53.7 | **42.8** | **25.7** |

**Grounding-DINO vs. YOLO-World.** To assess the influence of the open-set detector, we swap YOLO-World for Grounding-DINO inside MLFM (rows 1 vs. 6). Grounding-DINO ends up stronger for specific attribute types, such as $+8.0\%$ for *size*, $+28.5\%$ for *state* and $+1.8\%$ for *modifier*. Overall, we find that patch features are often better at capturing granular attribute cues than pixel or image features. However the choice of specific feature extractors and object detectors might heavily influence understanding of certain attribute types.

Table 4: On GOAT-Bench, MLFM+RGraph outperforms the baselines on language goals.

| Methods | trained | zero-shot | SR | SPL |
|---|---|---|---|---|
| RL Skill Chain | ✓ | × | 16.3 | 7.4 |
| RL Monolithic | ✓ | × | 12.6 | 6.5 |
| Modular GOAT | × | ✓ | 21.5 | 16.2 |
| MLFM+RGraph | × | ✓ | **24.2** | **21.5** |

**Computational and memory cost of 2D, multi-layer, and 3D maps.** To contextualize the efficiency of our representation, we compare the per-step cost of a 2D map (OneMap), our 3-layer multi-layer map (MLFM), and a 3D open-vocabulary map (VLMaps). MLFM introduces only a modest overhead relative to OneMap: the update takes $0.099 \pm 0.020s$ versus $0.083 \pm 0.018s$, and querying remains negligible ($0.00051 \pm 0.00030s$ vs. $0.00027 \pm 0.00012s$). In contrast, VLMaps is substantially slower, requiring $0.42 \pm 0.05s$ per update and $0.0018 \pm 0.00073s$ per query. Memory usage highlights the same trend. Our map covers a $60m \times 60m$ area at $6cm$ resolution ($1000 \times 1000$ grid). A dense 3D voxel grid with $1cm$ vertical resolution and a $2.5m$ ceiling would require storing a $250 \times 1000 \times 1000 \times 768$ tensor, which is prohibitive for real-time use. MLFM stores only three 2D layers, resulting in a compact $3 \times 1000 \times 1000 \times 768$ representation—almost two orders of magnitude smaller than a full 3D feature volume—while still retaining meaningful height cues.

## 5.4 RESULTS ON GOAT-BENCH

To demonstrate the effectiveness of our *MLFM+RGraph* method on real-world scans, we perform an experiment on the language goals from GOAT-Bench (Khanna et al., 2024b) and compare to three baselines from the GOAT-Bench paper – the zero-shot *Modular GOAT* (Chang et al., 2023) and two trained methods: *RL Skill Chain* and *RL Monolithic*. MLFM+RGraph, being a zero-shot method is however only comparable to the Modular GOAT method, that builds a semantic map memory (containing object categories) as well as an instance-specific map memory (containing egocentric images and CLIP image features). Table 4 shows that MLFM+RGraph outperforms the baselines on the language goals of GOAT-Bench, with +2.7% in SR and +5.3% in SPL.

## 6 CONCLUSION

We extend the semantic navigation task to *LaMoN*, to evaluate attribute-aware and spatially-aware language understanding. We propose *LangNav* dataset to implement this task, where episodes have sequences of goals, with manually verified descriptions and per-instruction linguistic tags. We also propose a novel method, *Multi-Layered Feature Map* (*MLFM*), that preserves fine-grained visual detail in a multi-layer semantic map representation and we demonstrate that it improves fine-grained and spatial language grounding. Experiments show that language understanding is still far from being solved and offer ample scope for new ideas to improve perception, memory, and language grounding. We hope our findings provide useful insights to drive future research in this direction.

**Limitations.** The linguistic descriptions in LangNav are restricted to phrases without complex structures such as coreference ("place *it* on the table"), negation ("a chair that is *not* black"), action directives ("pick up", "open the drawer"), or multi-step reasoning ("bring the mug you placed in the kitchen"). This design choice aims at isolating specific capabilities: *spatially grounded object identification* based on attributes and spatial relations. Incorporating richer phenomena such as reference resolution, manipulation planning, or temporally extended state tracking would introduce additional confounding factors, making it difficult to attribute performance differences to language grounding itself. Even within this controlled setting, our results show that current methods still struggle with fine-grained grounding (see Table 2), indicating that this core ability is far from solved. While more expressive linguistic constructs are important for broader home-service scenarios, we consider them orthogonal extensions and leave them for future work.

**LLM usage.** An LLM was used in the dataset generation process (see Section 3.1 and Appendix A.1.1 for detail), in the EgoImageMap+VLM baseline (see Section 4 and Appendix A.2.1) and in the MLFM+VLM method (see Section 4 and Appendix A.2.2). We also sparingly used an LLM (OpenAI, 2023) for rewording some text in the paper.

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

# A SUPPLEMENTARY MATERIAL

In this supplementary document we (i) present additional details of the LangNav corpus (Sec. A.1); (ii) provide more detail on the method (Sec. A.2); (iii) briefly describe the baselines used in the main paper (Sec. A.3); (iv) report further ablations that quantify the contribution of each component in our method (Sec. A.5); and (v) some considerations about real-world transfer (Sec. A.6).

## A.1 LANGNAV DATASET

In this section we present more details about our LangNav dataset, including details on dataset statistics (Table 5).

### A.1.1 LLM FOR DATASET GENERATION

**Forming coherent sentences from annotations.** We prompt GPT-4 (OpenAI, 2023) to form a sentence in natural language that would make sense to a human:

> "You will be provided with a list of attributes, and
> your task is to convert them to a free-flowing natural
> language sentence in English as spoken by native
> speakers. Drop brand names and all capital lettered
> words from the text."

We also provided a list of five examples to improve GPT's performance via in-context learning.

**Extracting attributes from sentences for annotations.** We use GPT-4 to also extract attributes from a language sentence which we then store as the fine-grained linguistic annotations, so that we can evaluate agent's performance on language understanding:

```
You are given a sentence with the description of
an object that someone is supposed to find in a
scene. Your goal is to extract the attributes used
to describe the object: <input_sentence>. Return the
output in a JSON format according to the following
format:

  {"attribute_types": {
  "color": {
      "exists": True if attribute is found in prompt or False
          otherwise,
      "explanation": list of attributes identified, or empty if
          none
  },
  "size": {
      "exists": True if attribute is found in prompt or False
          otherwise,
      "explanation": list of attributes identified, or empty if
          none
  },
  "shape": {
      "exists": True if attribute is found in prompt or False
          otherwise,
      "explanation": list of attributes identified, or empty if
          none
  },
  "number": {
      "exists": True if attribute is found in prompt or False
          otherwise,
      "explanation": list of attributes identified, or empty if
          none
  },
  "material": {
```

```
                    "exists": True if attribute is found in prompt or False
                        otherwise,
                    "explanation": list of attributes identified, or empty if
                        none
                },
                "texture": {
                    "exists": True if attribute is found in prompt or False
                        otherwise,
                    "explanation": list of attributes identified, or empty if
                        none
                },
                "function": {
                    "exists": True if attribute is found in prompt or False
                        otherwise,
                    "explanation": list of attributes identified, or empty if
                        none
                },
                "style": {
                    "exists": True if attribute is found in prompt or False
                        otherwise,
                    "explanation": list of attributes identified, or empty if
                        none
                },
                "text_label": {
                    "exists": True if attribute is found in prompt or False
                        otherwise,
                    "explanation": list of attributes identified, or empty if
                        none
                },
                "state": {
                    "exists": True if attribute is found in prompt or False
                        otherwise,
                    "explanation": list of attributes identified, or empty if
                        none
                },
                "modifier": {
                    "exists": True if attribute is found in prompt or False
                        otherwise,
                    "explanation": list of attributes identified, or empty if
                        none
                }}}
```

### A.1.2  EPISODE GENERATION

For each episode in the LangNav dataset, we first draw a random navigable start pose for the agent, then choose an object category that satisfies the following constraints: (i) at least one navigable viewpoint lies within 1.5 m of the object; (ii) the start pose and all three goals are on the same floor; (iii) every goal is reachable from the start pose; (iv) each goal is at least 2 m from the start pose; (v) the geodesic-to-Euclidean distance ratio from the start pose to each goal exceeds 1, ensuring non-trivial navigation; (vi) the geodesic distance from the start pose to the first goal is at least 2.5 m. For each goal, we also store 'viewpoints' or navigable positions around the goal object, such that navigating near to any of those will be considered successful (see Appendix A.1.5).

### A.1.3  SPATIAL RELATION DESCRIPTIONS

Here we describe in detail the procedure used to extract spatial relationships between a pair of objects from the ground-truth 3D object bounding boxes in HSSD. For *egocentric directional relations*, we take two objects with non-overlapping bounding boxes and compute the centroids of each bounding box. We transform them into the camera coordinate frame using the provided camera extrinsics. We then project these centroids into the image plane using the camera intrinsics and compare their horizontal pixel coordinates to assign "left" or "right" relations. This ensures that directional relations are defined relative to the agent's pose and viewpoint. For *allocentric relations*, we directly

Table 5: **Dataset statistics.** LangNav contains val and test splits, each with distinct sets of scenes and goal object categories. The goal descriptions might contain one or more linguistic features.

|                                          | Validation | Test | Overall |
| ---------------------------------------- | ---------: | ---: | ------: |
| scenes                                   | 20         | 15   | 35      |
| episodes                                 | 932        | 875  | 1807    |
| distinct object categories               | 19         | 12   | 31      |
| distinct object instances                | 160        | 154  | 314     |
| goal descriptions                        | 2796       | 2625 | 5421    |
| descriptions with attributes/relationships | 1659     | 2181 | 3840    |
| total linguistic features                | 2534       | 3524 | 6058    |
| unique linguistic feature values         | 136        | 99   | 235     |

compare the relative positions of object bounding boxes in the world coordinate system. For allocentric *directional relations*, we cast rays from the object's bounding box in the direction of the relation (vertical direction for "above" and "below") to retrieve the object with hits[2]. For instance, object $A$ is "above" $B$, if rays from $A$'s bounding box when shot downwards hit $B$'s bounding box. For allocentric *support relations*, we utilize the ground-truth support data available for the objects in HSSD. Allocentric *proximity relations* are derived by measuring the minimum Euclidean distance between the surfaces of two bounding boxes. We adopt a tiered thresholding scheme - objects within 0.0–0.2 meter are labeled as "next to" each other, while objects within 0.2–1.0 meter are labeled as "near" each other. This design captures both immediate adjacency and a more relaxed proximity. Finally, *containment relations* are identified when the bounding box of one object is fully enclosed within that of another across all three spatial dimensions. Our procedure benefits from the availability of ground-truth 3D bounding boxes in synthetic scenes, allowing us to generate accurate and consistent spatial relation annotations. The use of ray-casting provides a principled way to capture directional relations and the surface-to-surface distances enable a faithful encoding of adjacency to capture proximity relations. The pipeline is fully automatic and scalable, making it possible to construct a large dataset of spatial relations without requiring manual annotation.

### A.1.4 Understanding the dataset.

**Why the need for a new dataset?** The closest to our dataset is the GOAT-Bench (Khanna et al., 2024b) which introduced a multi-object navigation dataset that combines ObjectNav and InstanceNav goal descriptions. In contrast to GOAT-Bench, our dataset contains language descriptions at varying amounts of specificity, such that one or multiple target locations could be a match. We also include annotations for linguistic tags for each description which allows for fine-grained language evaluation. Moreover, GOAT-Bench contains several errors propagated from the VLM, which we mitigate by using ground-truth object annotations and also manually validating them.

The dataset creation pipeline in GOAT-Bench derives object attributes by prompting a pretrained BLIP-2 model, and this reliance on automatic extraction introduces noticeable noise in the captions. Typical issues include (Figure 4) ($i$) *partial matches*, where the instruction fits the target only in part; ($ii$) *hallucinations*, where attributes mentioned in the text are absent from the scene; and ($iii$) *mesh artefacts*, where incomplete geometry in HM3D-Sem hides the object, rendering the description misleading; ($iv$) *reference to object bounding box*, where the goal descriptions contain references to object bounding box, which the agent might not have access to during navigation (e.g. '...region defined by the bathtub bounding box'); ($v$) *spatial errors*, where the spatial relationship is wrong between objects (e.g. 'blanket near the bed on the left side' when it is clearly towards the foot of the bed in the figure). To obtain a statistic on how often these five error types occur, we randomly sample 100 goals from GOAT-Bench and manually inspect the descriptions along with the corresponding goal images. We found that only 33% of them were accurate whereas the rest had errors - 22% were hallucination errors, 7% were mesh artefact errors, 3% were partial match errors, 23% were spatial errors and 12% were due to reference to object bounding box or the image frame, totaling to

---

[2]Code adapted from Habitat-Lab https://github.com/facebookresearch/habitat-lab/blob/5e0d63838cf3f6c7008c9eed00610d556c46c1e3/habitat-lab/habitat/sims/habitat_simulator/sim_utilities.py#L724

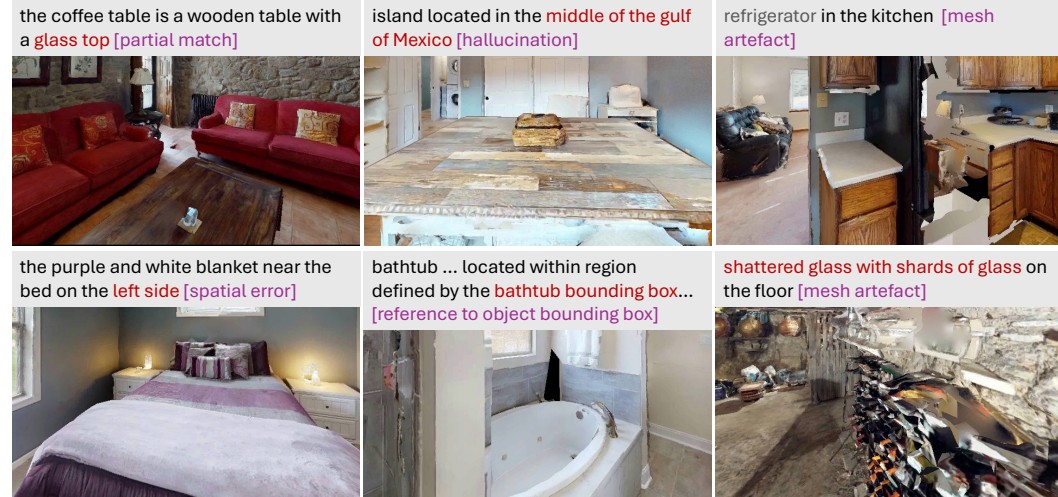

Figure 4: Language descriptions in Goat-Bench contain errors propagated from the BLIP-2 model. This figure shows examples for *partial match*, *hallucination*, *mesh artifact*, *spatial error* and *reference to object bounding box* errors.

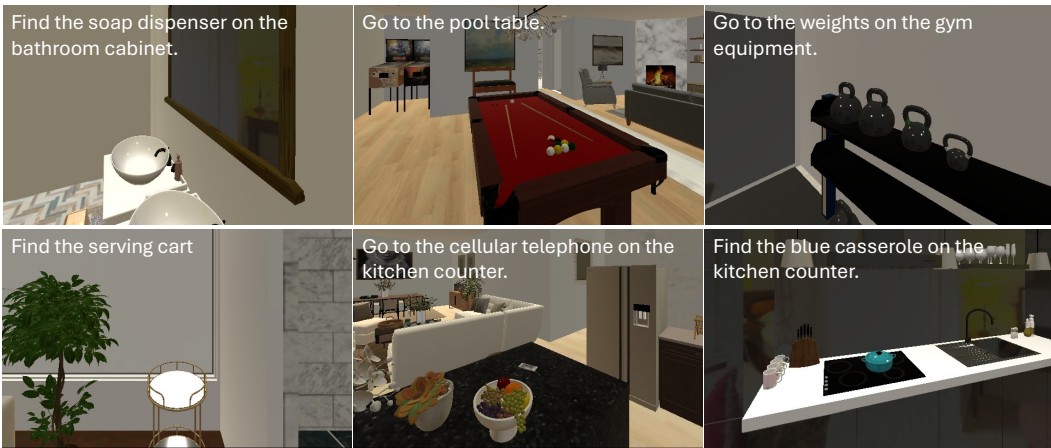

Figure 5: In LangNav, we use objects and attributes available in HSSD synthetic scenes, thus producing error free language descriptions.

67% error. The most frequent error types were spatial errors and hallucination errors. Moreover, in GOAT-Bench the language goals are generated to admit a single matching target, and they were not systematically validated to rule out additional scene objects that satisfy the same description.

We mitigated these errors by first using scenes and objects from HSSD dataset. The scenes are synthetic and hence free of mesh-related issues. The dataset also provides ground-truth categories and attributes for every object; we pass this structured metadata to GPT to obtain fluent natural-language instructions (Appendix A.1.1) that remain anchored to the ground truth. All generated sentences are then manually reviewed and, where necessary, corrected to eliminate any residual LLM errors (see Figure 5). Finally, we tag each instruction with the linguistic cues used in our benchmark (colour, size, material, state, and so on) (Appendix A.1.1) and include these annotations with the dataset—information that no previous semantic-navigation corpus offers. Moreover, all possible correct goal locations corresponding to a single description are added in the episode to account for multiple matches.

### A.1.5 VIEWPOINT GENERATION

Similar to the ObjectNav task (Batra et al., 2020), we store *viewpoints* or navigable positions around the goal object, such that navigating near to any of those will be considered successful. To generate candidate viewpoints around an object, we use depth observations together with the object's 3-D position and dimensions to obtain viewpoints with a clear line of sight to the target object. First, we trace the object's non-navigable boundary by sweeping a full $360^o$ and recording every transition point from free space to obstacle. Around each boundary point we then place radial samples at uniform intervals out to 1.5 m. Each sample is accepted as a valid viewpoint only if it meets two criteria: the location is navigable, and—after orienting the agent toward the object—no intervening obstacle blocks the view. We describe the full algorithm below.

---

**Algorithm 1** generate_view_points

---

**Require:** Input data `obj_pos`, `obj_dims`, $\text{dist}_{\text{bound}}$, $\text{radius}_{\text{bound}}$, $\text{dist}_{\text{vp}}$, $\text{radius}_{\text{vp}}$
**Ensure:** Output *Viewpoints around the object*
 1: Obtain `boundary_points` around the object
 2: **for** $\theta$ in $(0, 2\pi)$ **do**
 3:   Next equidistant $\text{pt}_{\text{bound}} \leftarrow$ Binary_Search ($\text{dist}_{\text{bound}}$, $\text{radius}_{\text{bound}}$, $\theta$ )
 4:   **if** $\text{pt}_{\text{bound}}$ exists **then**
 5:     Append $\text{pt}_{\text{bound}}$ to `boundary_points`
 6:   **end if**
 7: **end for**
 8: `view_pts` $\leftarrow \{\}$
 9: **for** $\text{pt}_{\text{bound}}$ in `boundary_points` **do**
10:   Compute equidistant `radial_pts` $\leftarrow f(\text{radius}_{\text{vp}}, \text{dist}_{\text{vp}})$
11:   **for** $\text{pt}_{\text{radial}}$ in `radial_pts` **do**
12:     Skip if within previous $\text{pt}_{\text{bound}}$ area
13:     Ensure $\text{pt}_{\text{radial}}$ is navigable
14:     Face the Agent towards `obj_pos` $\leftarrow f(\text{pt}_{\text{radial}}, \text{obj\_pos})$
15:     Compute corresponding pixel locations of `obj_dims`
16:     From *Depth* observation, obtain depth values at above pixel values
17:     **if** depth at $\text{pixel}_{\text{obj\_dims}} >$ distance to `obj_dims` **then**
18:       Append to `view_pts`
19:     **end if**
20:   **end for**
21: **end for**
22: **return** `view_pts`

---

## A.2 MLFM METHOD

### A.2.1 EGOIMAGEMAP+VLM BASELINE

For the EgoImageMap+VLM baseline method, we prompt a GPT-5 model with the following prompt to infer spatial relations between two objects in an egocentric RGB image:

```
"You are a vision-language assistant.  You are given
a query and a group of images.  Each group contains
several images.  For every image in each group, return
whether the object in the query is clearly visible.
Respond strictly in JSON. Group numbers are integers
(0, 1, 2, ...).  Use lowercase booleans ('true' /
'false').  Do not include explanations, only the
JSON."
```

### A.2.2 MLFM+VLM

In our MLFM+VLM variant, we prompt GPT-5 to infer the most likely relation between two objects from a list of images:

"Your task is to infer the most likely relation
between object A (colored in red) and object B
(colored in blue) from the list of images.

Note that the images are top-down map projections
and image 1 is physically at a lower height than
image 2 which is lower than image 3 and so on.  In
these images, you will find a green box, denoting the
agent location.  Return a json with "relation" and
"rationale" fields.

Follow these rules.  Use Euclidean proximity for
near/next to (next to = very close, non-overlapping;
near = close but not touching).  If green box for the
agent is present, decide LEFT/RIGHT relative to the
ray from agent toward the midpoint between A and B.
Consider the nearest blobs for A and B to the agent.
Be concise and avoid extra prose."

### A.2.3   MLFM+RGRAPH

In this section, we provide more detail into how we construct the MLFM+RGraph. The agent incrementally builds and updates the relation graph during navigation. As it encounters new views, nodes may be added or merged, and edges updated if new relational evidence is observed. Importantly, since objects may extend over multiple adjacent grid cells in the map, we collapse these into a single graph node and record the set of grid cells spanned by the object ($map\_span$). This aggregation serves two purposes: (1) it yields a segmentation-like mask of the object's spatial extent, and (2) it allows more accurate reasoning about its relation to nearby objects, which can now be computed relative to the object's full footprint rather than a single cell.

To infer relations between two nodes, we apply geometric rules on their $map\_span$, which records the grid cells occupied by each object as $[x, y, z]$ coordinates ($(x, y)$ for 2D grid location and $z$ for map layer). Vertical relations (above, below) are determined by comparing $z$ values, while proximity is inferred if objects are within $\sim$20 grid cells (grid resolution is 6 cm). Using the agent's pose, we transform object locations into the egocentric frame and their horizontal displacement then yields left or right relations. Containment is inferred by checking whether A's boundaries lie mostly within B's boundaries. We record all relations satisfying these rules for each node pair and store their corresponding text features in the edge. These geometric rules serve as a deterministic first step for relation inference; these can be replaced with a learned model trained on collected episodes to resolve ambiguous cases and support unseen relation types.

During the exploitation phase, the agent uses the RGraph to answer relational queries. Given a text query, we encode it using the CLIP text encoder and compute its cosine similarity with the stored graph features. Specifically, for each edge, we get (a) visual features for the source node, (b) visual features for the sink node and (c) the edge text features. The agent selects the candidate target node whose triplet score (target-relation-reference) is highest and exceeds a confidence threshold.

This formulation allows the agent to reason jointly over object semantics and spatial relations captured in the RGraph. Moreover, comparing CLIP text embeddings of the relational word allows the model to generalize to unseen but semantically similar relations (e.g., 'under' and 'below', 'within' and 'inside', 'beside' and 'next to'). In effect, RGraph provides a structured interpretable memory that grows with exploration and enables reasoning over spatial relations during exploitation.

### A.3   SEMANTIC NAVIGATION BASELINES

In this section, we briefly describe the baselines against which we compare our method in the main paper.

**VLMaps** (Huang et al., 2023) store LSeg (Li et al., 2022) pixel-level embeddings in a 2D top-down grid map, along with their 3D location. They average the pixel embeddings when multiple pixels are projected onto the same grid and also across multiple views.

Table 6: Varying map resolution and number of layers in our multi-layered map contributes towards MLFM's performance.

| | cm per 2D grid cell | Number of layers | SR↑ | SPL↑ |
|---|---|---|---|---|
| 1) | 6 | 1 | 29.7 | 9.3 |
| 2) | | 2 | 37.0 | 14.1 |
| 3) | | 3 | **39.5** | **14.9** |
| 4) | | 4 | 38.5 | 13.8 |
| 5) | | 5 | 27.6 | 10.1 |
| 6) | 10 | 3 | 25.1 | 9.6 |

**VLFM** (Yokoyama et al., 2023) builds a 2D value map that stores BLIP-2 (Li et al., 2023a) similarity scores of the observed images and object category, and uses it to semantically explore the environment. It achieves SOTA performance in the ObjectNav task.

**MOPA** (Raychaudhuri et al., 2024) iteratively builds a 2D category map after detecting objects in the frame, randomly explores the environment and navigates to the goal once found. We replace their PointNav (Wijmans et al., 2019) with the A* planner for fair comparison.

**OneMap** (Busch et al., 2025) builds a 2D feature map with SED (Xie et al., 2024) patch features by using a confidence-based fusion mechanism. They employ an open-set object detector YOLO-World (Cheng et al., 2024) along with the map to identify the target and use an A* planner to navigate.

### A.4 Additional Discussion on Texture and State Attributes

Both *texture* and *state* attributes show lower performance across models, but these results must be interpreted in light of their extreme sparsity in LangNav: only 1% of test episodes involve texture and 1.9% involve state. Such rarity makes the evaluation highly sensitive to a few difficult instances. Moreover, texture grounding is inherently challenging in zero-shot settings: fine-grained surface patterns are often lost during patch extraction, and current open-vocabulary encoders such as SED are not optimized for texture discrimination. While fine-tuning on texture-focused datasets could help, this falls outside our zero-shot evaluation protocol. State attributes present a different challenge: recognizing whether a door is open, or a mirror is illuminated, requires integrating global visual context. Patch-level features projected onto layered maps may fragment or obscure these cues, which explains why OneMap-v2, where all features lie on a single 2D plane, performs better specifically on state despite being inferior overall. Across both attributes, our analysis shows that the main limitations arise from the underlying open-vocabulary encoders rather than from the multi-layer map. Since MLFM relies exclusively on pre-trained detectors and visual encoders without any fine-tuning on LaMoN, the dataset effectively exposes current weaknesses in zero-shot perception models rather than shortcomings specific to our representation.

### A.5 Ablations

In this section, we report ablations to determine how the different components contribute to MLFM's performance. We ablate on grid resolution in Appendix A.5.1, followed by EAE percentage in Appendix A.5.2, and the contribution of map and object detector in Appendix A.5.3.

### A.5.1 Effect of Grid Resolution and Number of Layers

Table 6 reports an ablation on the two key map hyper-parameters: the number of height layers $L$ and the $(x, y)$ resolution of each slice. Success rises as $L$ grows from one to three layers, but degrades when a fourth layer is added. Beyond three slices large objects become fragmented, making them harder to localise; we therefore set $L = 3$ in all main experiments. Keeping $L$ fixed, performance improves when we refine the $(x, y)$-grid—i.e. when the centimetres-per-cell value decreases (rows 1 vs. 6). A finer resolution allows the map to preserve small objects' geometry and thus boosts the agent's success rate.

Table 7: The agent achieves the best performance when it spends 40% in Explore-and-Exploit (EAE) and the remaining time in Exploit-only phase.

| EAE (%) | SR↑ | SPL↑ | Error↓ | | |
|---|---|---|---|---|---|
| | | | Wrong detection | Wrong goal on map | OOT |
| 20 | 36.7 | 11.9 | **25.1** | 26.0 | **48.9** |
| 40 | **39.5** | **14.9** | 31.7 | 17.1 | 51.2 |
| 60 | 36.6 | 9.8 | 28.7 | 15.5 | 55.8 |
| 80 | 33.3 | 9.6 | 28.5 | **8.2** | 63.3 |

Table 8: MLFM achieves the best result when both the map and the object detector are used.

| Map | Object detector | SR↑ | SPL↑ |
|---|---|---|---|
| ✓ | × | 32.8 | 14.1 |
| × | ✓ | 29.0 | 8.6 |
| ✓ | ✓ | **39.5** | **14.9** |

### A.5.2 EFFECT OF EXPLORE-AND-EXPLOIT (EAE) PERCENTAGE

Table 7 varies the share of the episode spent to Explore-and-Exploit (EAE) versus Exploit-only (E). Overall performance peaks when the agent spends the first $40\%$ of its time budget in EAE and the remaining $60\%$ in E: both *Success Rate* ($SR$) and *Success weighted by Path Length* ($SPL$) achieve their highest values under this $40:60$ split. The table also reports the error composition. As the EAE portion grows, the *wrong-goal-on-map* rate declines, indicating that extended exploration yields a more accurate map. However, longer exploration leaves less budget for exploitation, so *out-of-time* failures increase once the EAE share exceeds $40\%$.

### A.5.3 ROLE OF MAP AND OBJECT DETECTOR IN MLFM

We next assess the individual contributions of the object detector and the multi-layer map. Table 8 compares three variants: map only, detector only, and the full system that combines both. Using both achieves the best performance followed by the *map-only* version followed by *detector-only* version.

### A.6 REAL-WORLD TRANSFER CONSIDERATIONS

In this section, we elaborate on the real-world relevance of our method and the empirical evidence supporting its transferability beyond synthetic environments.

**Modular architectures transfer more reliably than end-to-end policies.** Our approach follows a modular design consisting of two independent stages: (i) map construction from egocentric observations and (ii) path planning on the resulting map. This structure contrasts with fully end-to-end policies that directly map pixels to actions. Prior work has shown that modular pipelines exhibit far greater robustness in real-world deployment. Notably, Gervet et al. (2023) demonstrate that semantic-mapping agents achieve close to 90% success on real robots, while end-to-end trained agents performance drop moving from simulation to real environments. These findings suggest that mapping-based approaches such as MLFM are well aligned with the requirements of real-world deployment, as they rely on explicit scene representations that are more robust to the noise, distribution shift, and perceptual imperfections typical of physical environments.

**Use of real-image pre-trained perceptual encoders.** Although LaMoN is simulation-based benchmark, the perceptual stack used in MLFM consists entirely of off-the-shelf encoders and detectors trained on large-scale real-image datasets (e.g., BLIP-2, CLIP, SED), without any fine-tuning on HSSD. This design choice reduces the risk of overfitting to simulation-specific textures or meshes and contributes to improved robustness when transferring to real settings. In particular, the use of real pretrained encoders should make the feature representation significantly more stable under illumination changes.

**Evidence from cross-domain evaluation.** To probe generalization beyond LaMoN, we evaluated MLFM zero-shot on GOAT-Bench, a multi-modal navigation benchmark constructed from real RGB-D scans. GOAT-Bench introduces a number of challenges absent from LaMoN: BLIP-2 generated descriptions with hallucinations, mesh artifacts, incomplete rooms, partial occlusions, and realistic lighting. Despite these differences, MLFM achieved good zero-shot performance. The consistency between LaMoN and GOAT-Bench results suggests that the layered map representation and relational reasoning mechanism do not rely on simulation-specific regularities.

## B  POSSIBLE FUTURE WORK

The language in LangNav can be improved to include complex linguistic structures—coreference (e.g., "place *it* on the table") and negation ("a chair that is *not* black") and action directives beyond navigation verbs (e.g. "pick up", "bring back"). Our dataset can be further augmented with additional kinds of spatial relations such as egocentric proximity (e.g. "the chair near you"), allocentric relations based on object placement (e.g. "the chair in the corner of the room") or relational with respect other instances of the same objects (e.g. "the taller plant"). We leave this for future work. Our dataset generation approach makes a few assumptions. Bounding boxes only approximate true object geometry, which can lead to mislabels for irregularly shaped objects (e.g., a table placed in front of an L-shaped sofa may be incorrectly labeled as "inside" the sofa because its bounding box lies within that of the sofa). Fixed proximity thresholds (0.2 meter) may not generalize across scales (e.g., "near" may imply a few centimeters for tabletop objects but several meters in an office setting when describing "the water cooler near the meeting room"). Future work could incorporate more perceptually grounded measures, such as learned priors for spatial relations or scene-normalized distance metrics (e.g., defining "near" as within 10% of a room's longest dimension).

