# OpenReview forum: "MLFM: Multi-Layered Feature Maps for Richer Language Understanding in Zero-Shot Semantic Navigation"
_ICLR.cc/2026/Conference — Submitted to ICLR 2026_

### Official Review · Reviewer_WZtd · 2025-10-16

**Soundness:** 3
**Presentation:** 3
**Contribution:** 2
**Rating:** 6
**Confidence:** 3

**Summary:**

This paper introduces an open-vocabulary multi-object navigation dataset featuring natural language goal descriptions. It extends the traditional multi-object navigation task to a new setting, where an agent must locate a sequence of goals described in language. The authors further propose a novel approach that constructs a queryable, multi-layered semantic map using pretrained vision-language features, demonstrating its effectiveness in reasoning about fine-grained attributes and spatial relationships expressed in goal descriptions.

**Strengths:**

Overall, the problem setting is both realistic and practical, as it aligns closely with how users naturally specify sequential goals through language instructions.

**Weaknesses:**

In this setting, the task involves finding three sequential goals. I am curious about how this number was determined—have you experimented with different numbers of goals or tested the model’s performance under varying sequence lengths? What is the intuition behind choosing three? Additionally, it might be beneficial to provide a training dataset to further support the research community and enable more comprehensive exploration of this task.

**Questions:**

1. I am wondering whether the paper positions itself as an evaluation benchmark for open-vocabulary navigation, given that it provides only validation and test datasets.
2. In Section 4.2, the paper explains that MLFM+VLM is used to query relationships between objects, while MLFM+RGraph constructs a relational graph. Why isn’t the relationship queried directly from MLFM+RGraph? This seems like a chicken-and-egg problem—does the RGraph get constructed dynamically during the process?

---

> ### Author Response · Authors · 2025-11-22
> **Author Rebuttal**
>
> We thank Reviewer WZtd for the positive assessment. We appreciate the recognition that the proposed problem setting is both realistic and practical, closely reflecting how users naturally express sequential goals through language. This validation supports our aim of bridging evaluation benchmarks with real-world human–robot interaction patterns.
>
> > **Q1**: three sequential goals. I am curious about how this number was determined
>
> We selected a sequence length of three targets to follow the standard established in MultiON (Wani et al. 2020), which has become the common evaluation protocol for multi-object navigation. That said, the framework itself does not impose any restriction on the number of goals: since each environment contains multiple independent episodes, one can merge them or permute object lists to generate sequences of arbitrary length (e.g., 5, 10, or more targets). We will clarify this point in the revised manuscript.
>
>
> > **Q2**: I am wondering whether the paper positions itself as an evaluation benchmark for open-vocabulary navigation, given that it provides only validation and test datasets.
>
> Yes, the reviewer is right: LaMoN is designed primarily as an evaluation benchmark. As stated in the abstract, the goal is to enable systematic assessment of zero-shot agents or existing navigation models on fine-grained language grounding. The benchmark is not intended for training within LaMoN, but for testing whether agents trained elsewhere (e.g., on ObjNav, VLN-CE, or broader embodied-AI datasets) can correctly interpret spatial relations and attributes in multi-object scenarios. These settings require not only locating objects but also disambiguating between multiple candidates based on relational or attribute-level language (e.g., “the mug near the sink”), which is precisely the capability LaMoN aims to measure. Following the reviewer suggestion, we added a paragraph in section 3.1 in the revised paper: “evaluation focus”
>
> > **Q3**: In Section 4.2, the paper explains that MLFM+VLM is used to query relationships between objects, while MLFM+RGraph constructs a relational graph. Why isn’t the relationship queried directly from MLFM+RGraph? This seems like a chicken-and-egg problem—does the RGraph get constructed dynamically during the process?
>
> MLFM+VLM and MLFM+RGraph are two distinct query mechanisms built on top of the same multi-layered map. MLFM+VLM queries relations by prompting a vision–language model, while MLFM+RGraph never calls a VLM: it queries the dynamically constructed relational graph directly. The RGraph is indeed built online, step by step, as the agent updates the multi-layer map; at each step new nodes (objects) and edges (relations) are added or refined based on the current observations. The two variants therefore answer different questions: cosine-similarity matching on the layered map is effective for detecting object presence, whereas relational reasoning (e.g., “left of”, “on top of”, “near”) is naturally represented through edges in a relational graph. The RGraph is incrementally derived from the evolving map, and queried directly once constructed.

---

### Official Review · Reviewer_kpMR · 2025-10-25

**Soundness:** 3
**Presentation:** 3
**Contribution:** 3
**Rating:** 4
**Confidence:** 4

**Summary:**

This paper addresses the lack of language-centric evaluation in embodied semantic navigation by proposing three core contributions: (1) LangNav, an open-vocabulary multi-object navigation dataset with manually validated natural language goal descriptions and fine-grained linguistic annotations (attributes like color/size, spatial relations like "on/near"); (2) LaMoN, a task extending MultiON to require sequential navigation to language-specified goals with varying specificity; (3) MLFM, a multi-layered feature map method that balances 2D map efficiency and 3D height information, with three query variants (vanilla/VLM/RGraph) for fine-grained reasoning. Experiments on LangNav show MLFM+RGraph outperforms SOTA zero-shot baselines, and it generalizes to GOAT-Bench.

**Strengths:**

1. Manual validation eliminates VLM hallucinations (a major flaw of GOAT-Bench), and fine-grained tags enable dimensional evaluation of language understanding.

2. By requiring sequential language-specified goals without path instructions, LaMoN’s task better simulates real human-robot interaction than step-by-step VLN tasks.

3. The multi-layer map avoids 3D’s high memory cost and 2D’s height information loss.

**Weaknesses:**

1. Dataset limitations undermine generalizability: (1) Scenes are restricted to synthetic HSSD and semi-real GOAT-Bench. This paper did not test on physical environments (e.g., real apartments), where lighting/occlusions differ sharply from synthetic data. (2) Language is overly simple: no complex structures (coreference, negation) or action directives (e.g., "pick up"), making it irrelevant to real-world tasks like home service.

2. MLFM’s attribute understanding is incomplete: (1) Texture attribute success rate remains 0% (Table 2/3), but the paper only attributes it to feature extractors without proposing solutions or deeper analysis (e.g., fine-tuning CLIP on texture datasets). Moreover, state attribute (e.g., "illuminated mirror") underperforms OneMap-v2 (42.8% vs. 57.1%), but the failure analysis only simply mentions "detector issues" without detailed debugging. Better provide some experiments to discuss the issues mentioned above.

3. RGraph lacks adaptability: (1) Spatial relations are inferred via hand-crafted rules (e.g., Euclidean distance <0.2m = "next to"), not learned from data. This fails in scenes where "near" has different scales (e.g., 0.5m is "near" for a shelf but "far" for a table). (2) Cannot distinguish overlapping relations on the same layer (e.g., "inside" vs. "above" when two objects project to the same layer), with no workaround proposed.

4. Experiment scope is narrow: No comparison to state-of-the-art LLM-driven navigation methods (e.g., GPT-4V + embodied agents, 2024) – it’s unclear if MLFM is competitive with recent approaches. Besides, MLFM’s memory/compute cost vs. 2D/3D maps is unreported, which is critical for real-time robot deployment.

5. Some related and important works are missing citations: [1] NavQ: Learning a Q-Model for Foresighted Vision-and-Language Navigation [2] Learning Active Camera for Multi-Object Navigation [3] Mo-ddn: A coarse-to-fine attribute-based exploration agent for multi-object demand-driven navigation

**Questions:**

See weakness.

---

> ### Author Response · Authors · 2025-11-22
> **Author Rebuttal**
>
> We thank Reviewer kpMR for the insightful comments. We appreciate the recognition of LaMoN’s manually validated annotations and fine-grained tags, which address key limitations of prior benchmarks and enable more precise evaluation of language understanding. We also value the reviewer’s observation that LaMoN’s sequential, language-specified goals offer a more realistic interaction setting than step-wise VLN, and the acknowledgment that our multi-layer mapping strategy strikes a practical balance between 2D and 3D representations.
>
> > **Q1**: Dataset limitations undermine generalizability: (1) Scenes are restricted to synthetic HSSD and semi-real GOAT-Bench. This paper did not test on physical environments (e.g., real apartments), where lighting/occlusions differ sharply from synthetic data. (2) Language is overly simple: no complex structures (coreference, negation) or action directives (e.g., "pick up"), making it irrelevant to real-world tasks like home service.
>
>
> (1) Our method is fundamentally modular, consisting of two main stages: map building and path planning. Prior work, for example Gervet et al.’s “Navigating to Objects in the Real World”, shows that modular pipelines transfer far more reliably to the real world than end-to-end policies, with modular approaches achieving ~90% success on real robot deployments, whereas end-to-end methods drop from ~77% in simulation to ~23% in reality. In this light, our architecture is well aligned with real-robot deployment, because the map-building stage can reliably accumulate observations while the planning stage corrects residual mapping errors, errors that are far harder to correct in fully end-to-end systems. Regarding lighting, occlusions and other real-world nuisances, we agree they present non-trivial challenges; however, we wish to highlight that we use off-the-shelf feature extractors and object detectors pre-trained on large-scale real-image datasets rather than purely simulated images, not fine-tuned in any way on HSSD, and therefore we expect higher robustness to domain shift in the real-image domain. Moreover, our evaluation protocol included both synthetic scenes (from LaMoN) and real-world scenes (from GOAT-Bench) in a zero-shot setting (no fine-tuning on LaMoN or the real scenes), demonstrating cross-domain generalization. We added a dedicated discussion in the revised paper appendix (see section A.6).
>
> (2) The reviewer raises a valid point that we had already identified and explicitly stated in our Limitations section (L478–L482). The use of relatively simple language is a deliberate design choice aimed at isolating a specific capability: spatially grounded object identification based on attributes and spatial relations. Adding coreference, negation, or action directives (“pick up”, “open the drawer”) would introduce additional reasoning components, manipulation planning, task decomposition, state tracking, that are outside the scope of what we intend to evaluate here and would make it difficult to attribute performance differences to language grounding itself. Even within this controlled linguistic setting, our results show that current methods struggle with fine-grained grounding (see Table 2), which indicates that this core ability is not yet solved. While we acknowledge, as already stated in our submission, that richer linguistic phenomena are relevant for broader home-service scenarios, the present benchmark focuses on a well-defined and experimentally meaningful subproblem whose difficulty is evident from our evaluations. See the revised Limitations (part of Section 6)in the manuscript for reference.

---

> > ### Author Response · Authors · 2025-11-22
> > **Author Rebuttal**
> >
> > > **Q2**: MLFM’s attribute understanding is incomplete: (1) Texture attribute success rate remains 0% (Table 2/3), but the paper only attributes it to feature extractors without proposing solutions or deeper analysis (e.g., fine-tuning CLIP on texture datasets). Moreover, state attribute (e.g., "illuminated mirror") underperforms OneMap-v2 (42.8% vs. 57.1%), but the failure analysis only simply mentions "detector issues" without detailed debugging. Better provide some experiments to discuss the issues mentioned above.
> >
> > We carefully revisited the reviewer’s concerns regarding the attribute results (texture and state). Importantly, both attributes are extremely rare in the LaMoN test set: only 1% are texture episodes and 1.9% are state episodes. Such sparsity makes the evaluation highly sensitive to a few difficult cases and partially explains the instability of the reported numbers.
> > Texture: Texture-based references are intrinsically uncommon in real scenarios: users rarely specify objects by texture compared to more salient attributes such as color or category. The dataset distribution reflects this natural bias. In zero-shot conditions, texture is also inherently challenging: fine-grained surface patterns are easily lost when images are patchified to extract features, and generic vision–language encoders (e.g., SED) are not explicitly optimized for texture discrimination. Although fine-tuning on texture-specific datasets would likely help, it falls outside the zero-shot scope adopted in this paper. We will clarify this point in the Supplementary of the revised version.
> > State: The small number of state-related episodes makes performance fluctuate similarly. After inspecting failure cases, we found that patch-level features extracted with SED often lose the global context required to determine the state of an object. For example, whether a door is open or closed, or whether a mirror is illuminated, may be evident in the full image but becomes ambiguous once projected onto patches or split across layers. This also explains why OneMap-v2, where all features remain on a single 2D layer, performs better specifically on state attributes.
> > Overall, since our method is evaluated strictly zero-shot, using pre-trained detectors and visual encoders trained on real images (not on LaMoN), we view these results as exposing meaningful gaps in current open-vocabulary models rather than limitations specific to MLFM. Following the reviewer suggestion, we included these details and discussion in Appendix A.4
> >
> > > **Q3**: RGraph lacks adaptability: (1) Spatial relations are inferred via hand-crafted rules (e.g., Euclidean distance <0.2m = "next to"), not learned from data. This fails in scenes where "near" has different scales (e.g., 0.5m is "near" for a shelf but "far" for a table). (2) Cannot distinguish overlapping relations on the same layer (e.g., "inside" vs. "above" when two objects project to the same layer), with no workaround proposed.
> >
> > We agree with the reviewer that the current RGraph module relies on fixed, hand-crafted proximity thresholds, and we explicitly noted this in the paper (L1057–1058). A static threshold (e.g., 0.2 m) cannot fully capture differences in scale across environments; what counts as “near” for small tabletop objects differs from what is perceived as “near” in a large room. As we also discuss in the manuscript, a promising direction is to incorporate more perceptually grounded measures, such as learned relational priors or scene-normalized distance metrics (for instance, defining “near” relative to a fraction of the room size or to object dimensions). We will clarify these points more explicitly in the revised version. Regarding overlapping relations on the same layer, we acknowledge that this is a natural limitation of any 2.5D layer-based representation. One possible strategy would be to dynamically adjust the number or height of the horizontal slices. However, we believe this approach is not suitable in our setting for two reasons. First, similar to real-world scenarios, the agent only has access to the current request (e.g., “find the laptop on the desk”), and cannot anticipate future ones. Adapting the map structure based solely on the first query is therefore neither general nor reliable. Second, even if we were to regenerate a query-specific multi-layer map after exploration, this would require retaining and reprocessing all previous observations, computationally expensive and impractical for real-time operation, as it could involve storing and replaying hundreds or thousands of frames.

---

> > > ### Author Response · Authors · 2025-11-22
> > > **Author Rebuttal**
> > >
> > > > **Q4**: (1) Experiment scope is narrow: No comparison to state-of-the-art LLM-driven navigation methods (e.g., GPT-4V + embodied agents, 2024) – it’s unclear if MLFM is competitive with recent approaches. (2) Besides, MLFM’s memory/compute cost vs. 2D/3D maps is unreported, which is critical for real-time robot deployment.
> > >
> > > We would like to clarify that our evaluation already includes a strong VLM-driven baseline comparable to recent GPT-4V-style agents. The EgoImage+VLM baseline applies GPT-5 to egocentric RGB frames extracted from the simulator, and, as shown in Table 2, performs substantially below MLFM, indicating that purely vision–language approaches struggle when precise spatial grounding is required. Additionally, ollowing the reviewer feedback, we evaluated NaVid (RSS 2024) and UniNaVid (RSS 2025), two recent VLM-based navigation models fine-tuned across several embodied tasks (VLN-CE, ObjectNav, EQA, etc.).
> > > | Model         | SR         | SPL        | cat       | color     | size      | tex       | num       | mat       | state     | mod       | ego-dir   | allo-dir  | supp      | prox      | cont      |
> > > |---------------|-------------|-------------|------------|------------|------------|------------|------------|------------|------------|------------|------------|-------------|-------------|-------------|-------------|
> > > | **NaVid**    | 36.1  | 12.7  | 33.1 | 35.2 | 11.4 | 0.0  | 10.7 | 50.8 | 30.4 | 34.5 | 31.6 | 15.8 | 24.0 | 25.4 | 17.2 |
> > > | **Uni-NaVid**| **40.0** | 13.3 | **38.4** | **37.1** | 20.7 | **11.1** | 12.4 | **55.1** | **41.0** | **39.2** | 35.2 | 14.3 | 29.9 | 28.0 | 20.8 |
> > > | **MLFM+RGraph** | 39.5 | **14.9** | 33.6 | 34.9 | **41.7** | 0.0 | **14.3** | 53.8 | 14.3 | 37.6 | **37.7** | **20.7** | **34.2** | **32.5** | **21.1** |
> > >
> > > NaVid achieves lower performance (36.1% SR, 12.7 SPL), while UniNaVid reaches a Success Rate very close to MLFM+RGraph (40.0% vs. 39.5%). However, MLFM+RGraph retains a clear advantage in trajectory efficiency (SPL 14.9 vs. 13.3) and in attributes and spatial relations that require explicit spatial reasoning, such as number, support, containment, and egocentric direction, which are directly encoded in the multi-layer map. Conversely, UniNaVid performs better on categories dominated by detection quality, such as color, basic category, or texture, where differences mainly reflect the underlying detector rather than the mapping representation.
> > > These results are included in the revised PDF (please, see paragraph “Semantic and Vision and Language Navigation. “ in the related works and the updated Table 2 and the paragraph “Video-based VLM agents offer a complementary reference point.” in Section 5.1).
> > > However, we want to point out that Uni-NaVid, was trained for approximately 1400 GPU hours on 40 NVIDIA H800, and it is based on a 7 Billion parameter model, whereas MLFM is completely zero shot except for the visual detectors.
> > >
> > > Regarding compute and memory considerations, we thank you for your suggestion, for this reason we include a consolidated comparison across OneMap (2D), MLFM, and VLMaps (3D). Timing results show that MLFM introduces only a modest overhead relative to 2D maps while being much faster than VLMaps:
> > > – MLFM update cost: $0.099 \pm 0.020 s$, querying: $0.00051 \pm 0.00030 s$
> > > – OneMap: $0.083 \pm 0.018 s$, $0.00027 \pm 0.00012 s$
> > > – VLMaps: $0.42 \pm 0.05 s$, $0.0018 \pm 0.00073 s$
> > > Finally, in terms of memory usage, MLFM scales linearly with the number of vertical layers. Our map spans 60 m × 60 m at 6 cm resolution (1000 × 1000 grid). A dense voxel grid with 1 cm vertical resolution would require storing a $250 \times 1000 \times 1000 \times 768$ tensor (assuming a 2.5 m ceiling), which is computationally and memory-wise prohibitive. MLFM uses only 3 vertical layers, resulting in a manageable $3 \times 1000 \times 1000 \times 768$ representation, almost two orders of magnitude smaller than a dense 3D feature volume—while still retaining meaningful height information.
> > > We added a dedicated paragraph “Computational and memory cost of 2D, multi-layer, and 3D maps.” in section 5.3 of the revised paper.
> > >
> > > > **Q5**: Missing related citations
> > >
> > > Thank you for pointing this out. We integrated this missing related citations in the revised paper, see  paragraph “Semantic and Vision and Language Navigation. “ in the related works of the revised paper.

---

### Official Review · Reviewer_vF1j · 2025-11-01

**Soundness:** 3
**Presentation:** 3
**Contribution:** 3
**Rating:** 4
**Confidence:** 5

**Summary:**

This paper introduces LaMoN, a novel object navigation benchmark designed to address the limitations of existing ObjectNav benchmarks. It achieves this improvement primarily through two innovations: (1) It enriches the diversity of object descriptions by incorporating linguistic tags, allowing for more nuanced object definitions. (2) It integrates spatial relation descriptions for target objects, making the task more aligned with real-world navigation scenarios. These two features effectively narrow the gap between simulated benchmarks and practical applications, while also providing a more comprehensive evaluation platform for object navigation methods. Additionally, the paper proposes MLFM, a modular-based approach tailored for the LaMoN benchmark. Experimental results show that MLFM outperforms multiple baseline methods on ObjectNav tasks.

**Strengths:**

1. The proposed LaMoN benchmark is well-motivated with clearly presented details. Since most zero-shot object navigation (ZSON) approaches are still evaluated and compared on limited object categories and coarse-grained descriptions, this benchmark offers a superior alternative.
2. The paper presents a novel modular approach, MLFM, which is generalizable to different object navigation benchmarks including LaMoN and GOAT, and achieves competitive results on both.
3. The paper conducts detailed failure analysis and in-depth ablation studies on the impact of different types of image features in constructing the Multi-Layer Feature Map (MLFM) and the map query scheme.

**Weaknesses:**

1. The paper lacks essential comparisons with state-of-the-art navigation foundation models (e.g., NaViD[1], NaViLA[2], StreamVLN[3]) that are capable of handling general navigation tasks.

2. The contributions and differences between the proposed LaMoN benchmark and recent object navigation benchmarks (e.g., DOZE[4]) are not clearly justified.

3. The performance of MLFM in real-world scenarios remains unclear.

**Questions:**

1. What is the performance of recent end-to-end navigation foundation models on the LaMoN benchmark?

2. The paper presents the LangNav dataset—does this dataset benefit the training of navigation models on other object navigation benchmarks?

3. Given that obstacle sizes vary across different object types, how to define adaptive horizontal slices and construct a multi-layer feature map suitable for different scenes?

4. What is the decision frequency of the proposed MLFM approach?

---

> ### Author Response · Authors · 2025-11-22
> **Author Rebuttal**
>
> We thank Reviewer vF1j for the constructive feedback. We appreciate the recognition of LaMoN’s motivation and contribution as a more fine-grained and comprehensive benchmark for zero-shot object navigation, as well as the acknowledgement of MLFM’s generality across LaMoN and GOAT. We also value the reviewer’s appreciation of our detailed failure analysis and ablation studies, which were designed to provide a clearer understanding of the role of visual features and query mechanisms in MLFM.
> > **Q1**: What is the performance of recent end-to-end navigation foundation models on the LaMoN benchmark?
>
> Following this question, we ran additional experiments with NaVid, originally developed for VLN-CE where agents follow detailed route instructions rather than high-level exploration-driven goals, and with Uni-NaVid, a more general model trained across a broad set of embodied tasks (VLN-CE, Embodied QA, ObjectNav, etc.). These results are included in the revised PDF (please, see paragraph “Semantic and Vision and Language Navigation.” in the related works, the updated Table 2, and the paragraph “Video-based VLM agents offer a complementary reference point.” in Section 5.1).
>
>
> | Model         | SR         | SPL        | cat       | color     | size      | tex       | num       | mat       | state     | mod       | ego-dir   | allo-dir  | supp      | prox      | cont      |
> |---------------|-------------|-------------|------------|------------|------------|------------|------------|------------|------------|------------|------------|-------------|-------------|-------------|-------------|
> | **NaVid**    | 36.1  | 12.7  | 33.1 | 35.2 | 11.4 | 0.0  | 10.7 | 50.8 | 30.4 | 34.5 | 31.6 | 15.8 | 24.0 | 25.4 | 17.2 |
> | **Uni-NaVid**| **40.0** | 13.3 | **38.4** | **37.1** | 20.7 | **11.1** | 12.4 | **55.1** | **41.0** | **39.2** | 35.2 | 14.3 | 29.9 | 28.0 | 20.8 |
> | **MLFM+RGraph** | 39.5 | **14.9** | 33.6 | 34.9 | **41.7** | 0.0 | **14.3** | 53.8 | 14.3 | 37.6 | **37.7** | **20.7** | **34.2** | **32.5** | **21.1** |
>
> In terms of performance, NaVid achieves an SR of 36.1% and an SPL of 12.7, clearly below both Uni-NaVid and MLFM+RGraph. Uni-NaVid reaches an SR of 40.0%, very close to MLFM+RGraph (39.5%). However, MLFM+RGraph achieves a higher SPL (14.9 vs. 13.3), indicating that, even with similar success rates, our relationally structured multi-layer map supports more efficient trajectories.
> However, we want to point out that Uni-NaVid, was trained for approximately 1400 GPU hours on 40 NVIDIA H800, and it is based on a 7 Billion parameter model, whereas MLFM is completely zero shot except for the visual detectors..
> The two methods also differ in terms of fine-grained language grounding. MLFM+RGraph tends to perform better on attributes and relations that require spatial reasoning or are explicitly encoded in the multi-layer map, such as number, support, containment, and egocentric direction.
> Conversely, Uni-NaVid performs better on categories dominated by detection quality, such as color, basic category, or texture, where performance mainly depends on the underlying object detector rather than on the map representation itself.
> Finally, we note that MLFM brings a substantial improvement over standard 2D open-vocabulary maps under the same planner and detector. Success Rate increases from 26.7% with OneMap-v2 to 28.8% with MLFM-vanilla, and reaches 39.5% once relational-graph querying is added, an absolute gain of +12.8%. This confirms that the multi-layer representation and relational structure provide tangible benefits for fine-grained grounding and reliable navigation.
>
>
> > **Q2**: The paper presents the LangNav dataset—does this dataset benefit the training of navigation models on other object navigation benchmarks?
>
> In our experiments we focused on zero-shot VLM-based models that do not rely on task-specific fine-tuning, which allowed us to isolate the role of pre-training rather than introduce new sources of supervision. The results obtained with NaVid and Uni-NaVid suggest that models trained on data from related embodied navigation tasks generally benefit when transferred to LaMoN, supporting the idea that exposure to similar domains can improve generalization. At the same time, even these large pre-trained models exhibit unexpected weaknesses in some categories, for example, texture understanding remains particularly challenging. This reinforces our motivation for LaMoN: beyond serving as a dataset for our method, it provides the community with a focused benchmark for identifying where current navigation models transfer well and where substantial gaps remain.

---

> > ### Author Response · Authors · 2025-11-22
> > **Author Rebuttal**
> >
> > > **Q3**: Given that obstacle sizes vary across different object types, how to define adaptive horizontal slices and construct a multi-layer feature map suitable for different scenes?
> >
> > We appreciate the reviewer’s question. In our view, adaptive horizontal slices are not an ideal design choice for two main reasons. First, both in LaMoN and in real-world settings, the agent only has access to the current request (e.g., “find the laptop on the desk”) and not to future ones. As a consequence, dynamically tailoring the number or thickness of the layers to a single instruction is unlikely to produce a generalizable or consistent representation. Second, even if the agent were to accumulate observations over time and then rebuild a task-specific multi-layer map for each request, such a procedure would be computationally expensive and would slow down navigation significantly, especially in larger environments. For these reasons we opted for a fixed-layer design, which offers a stable decomposition of the scene and avoids introducing request-dependent variability into the map structure. As shown in Table 6, our ablation across different layer configurations indicates that a modest, fixed number of slices already provides a good balance between representational expressiveness and navigation efficiency. This suggests that most of the benefits of our approach arise not from adapting the vertical resolution to each scene, but from the multi-layered decomposition itself and from the relational reasoning applied on top of it.
> >
> > > **Q4**: What is the decision frequency of the proposed MLFM approach?
> >
> > We are not entirely certain about the precise meaning of “decision frequency” in the reviewer’s question, but if this refers to how often MLFM queries the map to decide whether the target has been found, then this occurs at every time step. Concretely, at each step the agent (i) updates the multi-layer map with the new observation and (ii) queries it to check whether the target object is now detected; if the target is not detected, the agent continues exploring following the navigation policy.
> >
> > > **Q5**: Real world concerns.
> >
> > Our method is fundamentally modular, consisting of two main stages: map building and path planning. Prior work, for example Gervet et al.’s “Navigating to Objects in the Real World”, shows that modular pipelines transfer far more reliably to the real world than end-to-end policies, with modular approaches achieving ~90% success on real robot deployments, whereas end-to-end methods drop from ~77% in simulation to ~23% in reality. In this light, our architecture is well aligned with real-robot deployment, because the map-building stage can reliably accumulate observations while the planning stage corrects residual mapping errors, errors that are far harder to correct in fully end-to-end systems. Regarding lighting, occlusions and other real-world nuisances, we agree they present non-trivial challenges; however, we wish to highlight that we use off-the-shelf feature extractors and object detectors pre-trained on large-scale real-image datasets rather than purely simulated images, not fine-tuned in any way on HSSD, and therefore we expect higher robustness to domain shift in the real-image domain. Moreover, our evaluation protocol included both synthetic scenes (from LaMoN) and real-world scenes (from GOAT-Bench) in a zero-shot setting (no fine-tuning on LaMoN or the real scenes), demonstrating cross-domain generalization. We added a dedicated discussion in the revised paper appendix (section A.6).

---

### Author Response · Authors · 2025-11-22
**General Remarks**

We thank all reviewers for their constructive feedback. We include all requested clarifications, experiments, citations and discussions into the revised version of the paper. All the new additions are in color text.
In particular we:
- Expanded the Related Work to include and discuss the suggested citations (NaVid, Uni-NaVid, NaVILA, StreamVLN, NavQ, MO-DDN, Learning Active Camera).
- Added new experiments evaluating NaVid and Uni-NaVid on LaMoN, now reported in Table 2 and discussed in Section 5.1, paragraph "Video-based VLM agents offer a complementary reference point".
- Included an analysis of computational and memory cost comparing 2D (OneMap), MLFM, and 3D map (VLMaps) representations (Section 5.3, paragraph "Computational and memory cost of 2D, multi-layer, and 3D maps.").
- Clarified the limitations of the dataset language in the revised Limitations subsection.
- Added a new appendix section discussing real-world transfer considerations (Appendix A.6).
- Expanded the discussion of texture and state attributes in Appendix A.4.

We believe these additions should address the main concerns raised by the reviewers and substantially improve the manuscript.  We are happy to answer any other questions and engage in further discussion with the reviewers.

---

### Comment · Area_Chair_zcfU · 2025-11-24

Dear Reviewers,

The authors have submitted their rebuttal addressing your concerns. Would you please kindly check the authors' responses and check whether your core concerns are adequately addressed, and adjust your initial score/comment if necessary? Your feedback is critical for the final decision. Thank you for your rigorous review and time!

Best regards, Your AC

---

### Author Response · Authors · 2025-12-02
**Follow-up on Rebuttal and Revisions**

Dear AC,

We thank the reviewers for their time and feedback. We have incorporated the reviewer suggestions into the paper, and we believe the manuscript has improved thanks to the comments from reviewers. We would greatly appreciate any follow-up questions or further suggestions you may have regarding our rebuttal or the revised version of the paper. We are happy to clarify any remaining concern, and would appreciate any feedback before the end of the discussion phase. Thank you again for your efforts.

---

### Meta-Review · Area_Chair_633L · 2026-01-03

**Summary:**

The paper received mixed initial reviews (4,4,6).

**What the paper contributes (as reviewers agree):**
1. **LangNav / LaMoN**
   - A language-centric evaluation benchmark for zero-shot semantic navigation, with fine-grained annotations for attributes (color, size, number, state, etc.) and spatial relations (on, near, containment, direction).
   - A new Language-guided Multi-Object Navigation (LaMoN) task that requires sequential navigation to language-specified goals without path instructions.
2. **MLFM**
   - A **multi-layered semantic map representation** that balances 2D map efficiency and 3D spatial awareness.
   - Multiple query mechanisms (vanilla, VLM-based, relational graph) for fine-grained grounding.

**Broadly acknowledged strengths:**
- The benchmark contribution is important and timely, addressing a gap in language-focused evaluation.
- Manual validation of annotations reduces hallucinations and enables interpretable analysis.
- MLFM outperforms prior zero-shot mapping baselines and generalizes to GOAT-Bench.
- Ablations, failure analysis, and post-rebuttal additions (new baselines, compute/memory analysis).

**Main axes of criticism:**
- Limited real-world realism (synthetic/semi-real scenes, simplified language).
- Incomplete attribute coverage, especially texture and state.
- Hand-crafted relational reasoning in RGraph.
- Initial lack of comparison to recent VLM-based navigation foundation models.
- Questions about scope (evaluation benchmark vs. training dataset).

**Reviewer Concerns:**

## Reviewer concerns that were addressed


**1. Compute and memory cost of MLFM:**
No quantitative evidence that MLFM is practical relative to 2D and 3D maps.

**Rebuttal:**
- Added a dedicated comparison of OneMap (2D), MLFM, and VLMaps (3D).
- Demonstrated modest overhead vs. 2D maps and large efficiency gains vs. 3D maps.


**2.  Clarification of benchmark scope (evaluation vs. training):**
Unclear whether LaMoN is intended for training.

**Rebuttal:**
- Explicitly clarified LaMoN as an evaluation-only benchmark.
- Added an “evaluation focus” paragraph to the paper.

**3. Justification of three sequential goals:**
Why exactly three goals?

**Rebuttal:**
- Explained alignment with the **MultiON standard**.
- Clarified that the framework supports arbitrary sequence lengths.

**4. Adaptive slicing and decision frequency:**
How are layers chosen, and how often is the map queried?

**Rebuttal:**
- Provided a principled argument for fixed layers.
- Clarified that map update and query occur at every timestep.



**5. Missing related work citations:**
Several important navigation works were missing.

**Rebuttal:**
- Added all requested citations and expanded related work.



##  Reviewer concerns that remain partially outstanding

**1. Missing comparisons to navigation foundation models:**
No evaluation against recent VLM-based navigation agents (NaVid, Uni-NaVid, GPT-style agents).

**Rebuttal:**
- Added NaVid (RSS’24) and Uni-NaVid (RSS’25) results on LaMoN.
- Showed comparable SR to Uni-NaVid, higher SPL, and clear gains on spatially grounded attributes.
- Clarified that MLFM is fully zero-shot, while Uni-NaVid is a large, heavily trained model.


**2. Limited realism of language and environments**
- Language lacks coreference, negation, and action directives.
- Environments are synthetic or semi-real.

**Rebuttal:**
Acknowledged and justified as a deliberate scoping choice.



**3. Weak performance on texture and state attributes**
- Texture success rate ≈ 0%.
- State attributes underperform 2D baselines.

**Rebuttal:**
Explained via data sparsity and feature limitations, but no concrete mitigation proposed.



**4. Hand-crafted relational reasoning in RGraph**
- Fixed thresholds for spatial relations.
- Ambiguity when relations overlap on the same layer.

**Rebuttal:**
Explicitly acknowledged; future work suggested.

**Reviewer Scores:**

This paper initially received scores 4, 4, and 6. The negative scores seem hard to raise to or above 6 after the rebuttal due to the partially outstanding concerns listed above. For example, the lack of comparisons with broad recent language navigation methods (even with the newly added comparisons against NaVid, Uni-NaVid, etc) is far from enough. This is a major issue for a benchmark work.

---

### Decision · Program_Chairs · 2026-01-26

Reject